# Differences in Cannabis and Cannabidiol Attitudes, Perceptions, and Behaviors Between US Adolescents Receiving Mood Disorder Treatment and Their Parents Across Legal Contexts

**DOI:** 10.3390/ijerph22101576

**Published:** 2025-10-16

**Authors:** Christopher J. Hammond, Mary A. Fristad, Yoon Ji Moon, Melissa M. Batt, Richard Dopp, Neera Ghaziuddin, Leslie Hulvershorn, Jarrod M. Leffler, Manpreet K. Singh, Aimee E. Sullivan, Sally Weinstein, Leslie Miller

**Affiliations:** 1Department of Psychiatry & Behavioral Sciences, Johns Hopkins University School of Medicine, Baltimore, MD 21224, USA; lmille84@jhmi.edu; 2Department of Pediatrics, Johns Hopkins University School of Medicine, Baltimore, MD 21224, USA; 3Division of Child & Family Psychiatry, Nationwide Children’s Hospital and The Ohio State University, Columbus, OH 43205, USA; 4Division of Neurology, Children’s Hospital of Philadelphia, Philadelphia, PA 19104, USA; 5Department of Psychiatry, University of Colorado Anschutz Medical Campus, Aurora, CO 80045, USA; 6Department of Psychiatry, University of Michigan, Ann Arbor, MI 48331, USA; dopp@med.umich.edu (R.D.);; 7Department of Psychiatry, Indiana University School of Medicine, Indianapolis, IN 46202, USA; 8Department of Psychiatry & Psychology, School of Medicine, Virginia Commonwealth University, Richmond, VA 23298, USA; 9Department of Psychiatry & Behavioral Sciences, University of California Davis, Sacramento, CA 95817, USA; 10Department of Psychiatry, College of Medicine, University of Illinois at Chicago, Chicago, IL 60612, USA; sweins3@uic.edu

**Keywords:** cannabis, cannabidiol, adolescents, parents, mood disorders, attitudes, perceptions, expectancies, cannabis use intentions

## Abstract

Dramatic shifts in state-level cannabis laws (CLs) and federal hemp regulations have resulted in increased availability and use of cannabis and cannabidiol (CBD) products throughout the US, with unknown implications for the youth. Youth with mood disorders represent a vulnerable population that is more likely to use cannabis and CBD and is at elevated risk for experiencing cannabis-related adverse health outcomes. This multisite study characterized attitudes, health perceptions, and behaviors related to cannabis and CBD use among US youth receiving mood disorder treatment and their parents, and assessed whether attitudinal differences varied as a function of respondent group and state-level CL status. Anonymous surveys were completed by 84 youths and 66 parents recruited from six child mood clinics providing care to patients living in eleven US states with variable CLs. Covariate-adjusted regressions were run using respondent group and state-level CL status as between-subject factors. Most youths (76% and 74%) and parents (65% and 68%) endorsed believing that cannabis and CBD, respectively, are safe and effective treatments for mental health conditions, and that regular use of these products reduces depression, anxiety, and suicidal behaviors. Intergenerational differences in cannabis-related attitudes and health perceptions were observed, with some associations varying as a function of state-level CL. Among the youth, male sex and positive cannabis expectancies and attitudes were associated with higher cannabis use intentions. Findings can inform prevention and public health messaging efforts.

## 1. Introduction

Rapid changes to United States (US) state-level cannabis laws (CLs) over the past 20 years carry unknown implications for youth health. As of November 2024, medical and recreational cannabis are legal in 39 and 24 states, and Washington, DC, respectively [1]. State-level legislative changes to cannabis’s legal status and federal passage of the 2018 Farm Bill deregulating the production and sale of hemp, a cannabis plant variant containing < 0.3% of delta-9-tetrahydrocannabinol (delta-9-THC) [2], have led to a billion-dollar US cannabinoid market, resulting in increased availability and use of cannabis-derived and hemp-derived cannabinoid products across the US [2]. The main plant-derived cannabinoids from cannabis are delta-9-THC, the main psychoactive constituent responsible for intoxicating effects, and cannabidiol (CBD), a non-psychoactive cannabinoid under investigation for therapeutic properties [3]. In the US, CBD is available as a prescribed medicine that is approved by the US Food & Drug Administration (FDA) for two rare seizure disorders (Lennox–Gastaut and Dravet Syndrome) in children and tuberous sclerosis in adults; in some cannabis products obtained from medical cannabis (MC) dispensaries and retail outlets in states with CL; and in hemp-derived dietary and health supplements in multiple formulations (e.g., gummies, vapes, tinctures, CBD-infused food, drinks, and topicals) [4]. Cannabis (used medically or recreationally) and hemp-derived CBD supplements are the most commonly used cannabinoid-based products in the US. There has been growing demand for cannabis and CBD across the US amidst broad claims of purported health benefits that have greatly outpaced the scientific literature substantiating their use as therapeutic agents [5]. Downstream public health effects related to shifting medical cannabis laws (MCLs) and recreational cannabis laws (RCLs) and the emerging cannabinoid market are unclear and may have both negative and positive outcomes for different subgroups [6,7].

Youth undergo rapid neurodevelopmental changes, making them more prone to experimentation with cannabis and making their brains more sensitive to cannabis exposure effects compared to adults [8]. This, in combination with the high rates of youth cannabis use [9], makes them a potentially vulnerable subgroup with regard to negative health outcomes related to changing state and federal cannabis policies and expanded societal availability [6,7,10,11,12,13,14]. Psychiatric disorders, particularly mood, psychotic, and substance use disorders, can be exacerbated by regular cannabis use, and youth with psychiatric disorders may be more vulnerable to cannabis-related adverse health outcomes, including elevated risk of substance use disorders and poorer course of serious mental illness [6,10,11,12,13,15]. In the current cannabis policy landscape, youth with mood disorders represent a uniquely vulnerable population that is both more likely to use cannabis and is at elevated risk for experiencing cannabis-related adverse health outcomes [6,16,17]. At the same time, rates of CBD use have increased in US youth, and may be higher in youth with mood problems [18,19]. Given this, research into cannabis and CBD attitudes, medical beliefs, and perceptions, as well as the influence of changing CLs on cannabis and CBD use in youth with mood disorders, is warranted [20].

Cannabis remains the most commonly used federally illicit drug by US adolescents and young adults, with cannabis use disorder being the most common substance use disorder for which US adolescents receive treatment [6]. Over the past two decades, cannabis-containing products have become more potent and available, and the perception of risk for harm-related cannabis use among US youth, in general, has decreased, resulting in a more permissive environment and increased access to cannabis products with high potency/concentrations of delta-9-THC [6,21]. This is concerning given that use of cannabis, particularly high delta-9-THC potency cannabis, is associated with adverse health outcomes, increased risk for psychiatric disorders, academic failure, and higher rates of morbidity and mortality [22,23,24]. CBD use has also increased among US youth and adults, driven primarily by increased use of CBD-containing dietary/health supplements [20]. A recent national survey showed that 19.8% of US 16–25 year olds reported using CBD supplements in the past year [25]. Rates of vaping CBD and using CBD supplements to target health concerns have also increased among US youth [18]. Many US adults and youth report using CBD supplements for health concerns (often without consulting their doctor) in an attempt to self-medicate mood symptoms, pain, or sleep problems or to replace their conventional medications, despite limited evidence of safety and efficacy [19,25,26,27]. This is also concerning, as a major unanswered question in the field remains whether CBD exposure during critical periods disrupts neurodevelopmental processes in a way similar to delta-9-THC (see Ref. [28] for review). While generally considered to have a good drug safety profile, CBD products are not risk free. The use of CBD products may pose risks for side effects, adverse events, and supplement-to-drug interactions with prescribed medications in some patients [29]. Furthermore, several studies have shown that commonly available hemp-derived CBD supplements are often mislabeled, contaminated, or adulterated, and may contain trace amounts of THC in sufficient doses to produce psychoactive effects or result in positive urine drug test results for cannabis [30] that could impact health, employment status, and sports participation [31]. Thus, the risks related to CBD use may be amplified, and the potential benefits may be more difficult to measure for patients who use CBD supplements compared to pharmaceutical CBD medications [5]. In light of this, studies investigating real-world use of different cannabis chemotypes and CBD products in youth are needed.

Cannabis and CBD-related attitudes, expectancies, and health perceptions influence cannabis use intentions and behaviors [26,32,33], which may be influenced by changing CLs [20]. Health perceptions about medical benefits and risk for harm related to cannabis use also influence cannabis use intentions and behaviors [34,35]. Cannabis-related risk perceptions have declined across all US age groups, with greater declines found among individuals living in states that have enacted RCL and MCL [36,37,38]. RCL is associated with increased perception of health benefits related to cannabis for pain, stress, and anxiety [39]. Changing state-level CLs may also affect parent attitudes, expectancies, health perceptions, and parenting behaviors related to cannabis and CBD use. After RCL was passed in Washington, parental attitudes became more favorable for adult use, and perception of the harmfulness of cannabis substantially decreased, while attitudes about adolescent use were mixed [40]. National surveys indicate cannabis use has increased among US adults who have children in the home and is higher in states that have legalized cannabis [41].

This article presents findings from a multisite/multistate study of US youth receiving mood disorder treatment and their parents who were recruited from child mood disorder clinics affiliated with the National Network of Depression Centers (NNDC) across the US, in states with variable CLs. Via anonymous surveys, aims of the current report were to (1) characterize attitudes, expectancies, health perceptions, and use intentions related to cannabis and CBD use among youth and parents; (2) test for intergenerational differences; (3) examine the influence of state-level CLs (i.e., RCL vs. MCL vs. No-CL) on cannabis-related and CBD-related attitudes, expectancies, health perceptions, and behaviors; and (4) identify whether individual-level factors (e.g., demographic and clinical characteristics as well as personal attitude/expectancies), and state-level CL status were associated with youth intention to use cannabis and CBD.

## 2. Materials and Methods

### 2.1. Study Overview

This report used youth and parent data from an anonymous online survey developed and administered to youth receiving mood disorder treatment in the US, their parents, and mental health providers as part of a larger multisite cross-sectional study examining cannabis and CBD-related attitudes, beliefs, and behaviors across states with different CLs. Research procedures were approved by each site’s Institutional Review Board (IRB). The Strengthening the Reporting of Observational Studies in Epidemiology (STROBE) reporting guideline for cross-sectional studies was used [42].

### 2.2. Participants and Procedures

#### 2.2.1. Participants

Adolescents (ages 12–17) and young adults (ages 18–25) receiving treatment at youth mood disorder clinics in six states and parents/caregivers (henceforth referred to as parents) of youth receiving mood disorder treatment were invited to complete an anonymous survey about cannabis and CBD-related attitudes and behaviors.

#### 2.2.2. Study Eligibility

For youth, inclusion criteria included ages 12–25 years and currently receiving mental health treatment at one of the six participating youth mood disorder clinics. For parents, inclusion criteria included being a primary caregiver of an adolescent who was currently receiving mental health treatment at the clinics. Exclusion criteria for both respondent groups included lack of English language fluency and cognitive deficits or acute severe mental illness that would interfere with their ability to safely participate in study procedures, understand item content, and complete the survey. Parent and youth participants were not limited to parent–child dyads. Broad inclusion criteria and few exclusion criteria were used to optimize generalizability by allowing participation by a diverse patient population representative of real-world patients from youth mood clinic settings.

#### 2.2.3. Recruitment

Participants were recruited from NNDC-affiliated youth mood disorder clinics in six states with variable state-level CLs. Sites were geographically distributed, included urban and rural populations, and represented the US Mid-Atlantic, North Central, and Mountain regions.

#### 2.2.4. Procedures

Consent was obtained from participants ≥18 years, and parental consent and assent were obtained from participants < 18 years. Recruitment and study procedures were the same across sites, with data collection conducted by local study team members. Participation was optional, and a $10 gift card was provided as compensation. Surveys were administered online using REDCap software; one site used paper surveys. Data collection occurred from July 2020 to March 2023.

### 2.3. Measures

#### 2.3.1. Marijuana Attitudes, Beliefs, and Behaviors Study (MABS) Survey Instrument

The MABS survey was developed for this study from themes elicited from group discussions with child mood disorder experts at NNDC Child and Adolescent Mood Disorder Interest Group meetings, focused on better understanding youth, parent, and provider attitudes, beliefs, and behaviors related to cannabis and CBD in the current US healthcare environment and cannabis policy landscape. Survey development was guided by expert discussion around clinically relevant themes for youth mood disorders and informed by a scoping literature review on the topic, with items chosen or developed to address existing gaps in the literature. Many survey items were modeled after those used in previous survey studies [40,43,44,45], with some taken directly from validated instruments (e.g., risk perception items). The full survey included 129 items querying cannabis- and CBD-related attitudes and beliefs (5 items); use intentions (3 items); health perception of risk for harm and medical benefit (4 items); perceived accessibility (1 item); perceived use norms (1 item); perceived knowledge about cannabis and CBD use effects (12 items); MC and CBD use behaviors (4 items); expectancies about cannabis, MC, and CBD’s effects on mood, anxiety, and other psychiatric symptom domains (48 items); parent–youth communication and parenting practices (31 items); mental health provider practice behaviors and patient–provider communication about cannabis, MC, and CBD (11 items); demographics (5 items); and clinical characteristics (6 items). The survey used branching logic. After completing demographic and clinical questions, participants received definitions for MC and CBD and were asked if they had heard of these products. Participants who had not heard of MC or CBD were instructed to skip related survey items. Parallel youth, parent, and provider versions of the survey were created. See Appendix A for additional details.

Items from the MABS survey used as variables in the current report included the following: demographics (age, sex, parent education, and state of residence), clinical/treatment characteristics (lifetime psychiatric symptoms from different domains/categories and current treatment type/setting/modality), cannabis and CBD medical beliefs and attitudes, perceptions about risk for harm and benefits for mental health from regular use of cannabis and CBD, as well as cannabis and CBD use intentions and behaviors.

#### 2.3.2. Marijuana Effect Expectancy Questionnaire—Brief (MEEQ-B)

The MEEQ-B, a six-item validated questionnaire, measures individual cannabis expectancies across six domains (cognitive and behavioral impairment, relaxation/tension reduction, social facilitation, sexual facilitation, cognitive/perceptual enhancement, global negative effects, and craving/physical effects). It includes positive and negative effect expectancy subscales and has been validated in adolescent and adult samples [45].

#### 2.3.3. Patient Health Questionnaire—2-Item Version Depression Scale (PHQ-2)

The PHQ-2 [46] is a brief depression measure that includes the first two items from the PHQ-9 and assesses the frequency of depressive symptoms in the past 2 weeks. It has demonstrated validity in adolescent populations [47]. PHQ-2 scores range from 0 to 6, with a cutoff of ≥3 representing an optimally sensitive/specific cutpoint for identifying individuals with a current depressive disorder.

#### 2.3.4. Generalized Anxiety Disorder—2-Item Anxiety Scale (GAD-2)

The GAD-2 is a brief anxiety measure that includes the first two items from the GAD-7 and assesses the frequency of anxiety symptoms in the past 2 weeks [48]. It has demonstrated validity in adolescent populations [49]. GAD-2 scores range from 0 to 6, with a cutoff of ≥3 representing an optimally sensitive/specific cutpoint for identifying individuals with a current anxiety disorder.

MEEQ-B positive and negative expectancy scores and PHQ-2 and GAD-2 scores were used as continuous measures of cannabis expectancy, depression, and anxiety in the analyses.

### 2.4. State-Level Cannabis Laws

Using US National Conference of State Legislatures data [1], respondent-reported state of residence was linked with state-level CLs and coded into one of three hierarchical categories: (1) RCL; (2) MCL; and (3) No-CL. All RCL states previously enacted MCL, which remained active; no states changed CL status during data collection. See Appendix A.

### 2.5. Statistical Analysis

We used IBM SPSS Statistical Analysis Software v25.0 (IBM, Armonk, NY, USA); significance was set at two-sided *p* < 0.05. Discrete missing values were minimal (0% to 3.0%) and imputed using mean substitution methods. Results were summarized using descriptive statistics to characterize beliefs, attitudes, perceptions, and expectancies for cannabis and CBD in the total sample and across respondent groups. Multinomial logistic and linear regression analyses with state-level CL status (RCL vs. MCL vs. No-CL) and respondent groups (youth vs. parent) as between-group factors tested for state-level CL and intergenerational differences. We ran separate post hoc covariate-adjusted models for group and CL status effects in the total sample, then examined CL status effects separately for youth and parents to parse group-by-CL-status interactions. Age, sex, parent education, and site were included as covariates of no interest in the models. We used a two-step analysis to investigate demographic, clinical, attitudinal, and state CL factors that were associated with intentions to use cannabis and CBD in the youth respondent group. As a first step, to reduce data, we ran ordinal regression models for each factor, covarying age, sex, parent education, and site. Then, we used multivariable ordinal regression analyses to characterize predictors of use intentions, incorporating each factor that significantly correlated with cannabis and CBD use intentions to determine whether those factors were associated with intentions to use after controlling for other relevant variables.

### 2.6. Sensitivity Analyses

Four sensitivity analyses were performed to ensure robustness of the main results and to test whether differences related to group and CL status varied as a function of other measured and unmeasured factors. First, we applied leave-one-out validation (LOOV), holding out/censoring respondents from each site and running main analytic models in the *n*-1 sample to ensure results were not being driven by a single site. Second, we reran analyses in a subgroup of matched youth–parent dyads (30 respondents/group). This test enabled us to examine differences in beliefs, attitudes, perceptions, and expectancies in relation to respondent group and CL status in a sample with limited between-group variation from measured and unmeasured environmental factors. Third, we reran analyses in the youths with current depression and anxiety disorders (i.e., PHQ-2 or GAD-2 ≥ 3) to determine whether the impact of current depression or anxiety, meeting clinical/diagnostic thresholds, influenced the associations identified in our main analyses. Finally, we reran analyses for youth with no lifetime history of alcohol or drug use problems (75 youth and 57 parent reports).

## 3. Results

### 3.1. Sample Characteristics

We enrolled 156 individuals; 150 respondents (96.2%) who completed > 75% of items (84 youth, 66 parents; see Table 1) make up the final analytic sample for the study, representing a convenience sample of youth receiving mood disorder treatment in the US and their parents. Youth and parent respondents from the analytic sample were similar regarding sex and state CL status but differed on age, parent education, and total psychiatric problem scores.

Most respondents had heard of MC (95.3%) and CBD (92.0%). Those who had not were predominantly ≤15 years old (71% and 83%, respectively) and younger than their counterparts (Mean age = 20.0 vs. 31.7 years; t = 2.55, *p* = 0.012), but did not differ on sex, parent education, depression severity, anxiety severity, presenting clinical characteristics, or state CL.

Participants were from eleven US states (six states with clinic sites, five adjacent states) with variable CL (three RCL [*n* = 63, 42%]; four MCL [*n* = 51, 34%]; four No-CL [*n* = 36, 24%]). Respondents from states with different CLs did not differ in age, sex, depression or anxiety severity, or presenting clinical characteristics. Parents from RCL states (vs. MCL and No-CL) had higher educational levels (“some college/university”: 70.5% vs. 49.0% vs. 58.3%, respectively).

### 3.2. Medical Beliefs, Expectancies, and Behaviors

Most youth endorsed believing that cannabis (76.2%) and CBD (74.3%) are safe, effective treatments for certain mental health conditions. Approximately half agreed that “mental health providers should be recommending or prescribing cannabis (48.8%) and/or CBD (58.1%)”. Many endorsed that regular use of cannabis (CAN) or CBD improves depression (CAN: 54.8% and CBD: 50.0%), anxiety (CAN: 59.5% and CBD: 63.5%), suicidal thoughts/behaviors (CAN: 44.0% and CBD: 32.4%), and psychotic symptoms/disorders (CAN: 20.3% and CBD: 27.1%).

Most parents endorsed that cannabis (65.2%) and CBD (68.8%) are safe, effective treatments for adults (depression, CAN: 53.0% and CBD: 56.3%; anxiety, CAN: 60.6% and CBD: 62.5%; suicidal thoughts/behaviors, CAN: 33.4% and CBD: 29.7%; and psychotic symptoms, CAN: 18.2% and CBD: 17.2%) and agreed that “mental health providers should be recommending or prescribing cannabis (50.0%) or CBD (54.7%)”. Fewer thought cannabis (28.8%) and CBD (40.9%) were safe and effective for youth. A sizeable minority endorsed that mental health providers should prescribe or recommend cannabis (25.8%) and CBD (34.8%) for pediatric mental health conditions.

A minority of youth (*n* = 11, 14.9%) and parents (*n* = 10, 15.6%) reported using CBD to treat their mental health condition in the past year; 17.2% of parents reported that their child had done so. Slightly more (21.6% youth and 22.2% parents) reported a household member had used CBD in the past year. In those who reported any CBD use (individual or household), 41.9% used hemp-derived products, 53.5% cannabis-derived products, and 27.3% did not know the source. CBD was used to treat anxiety (76.7%), depression (48.8%), inattention/hyperactivity (37.2%), anger/irritability (25.6%), pain (18.6%), and sleep problems (16.3%). Most (74.4%) reported that CBD was somewhat/very helpful.

Among respondents living in MCL or RCL states (*n* = 114), 8.7% of parents reported past-year MC use to treat a mental health condition; 8.8% of youth and 10.9% of parents reported past-year MC use by a household member.

### 3.3. Youth Versus Parent Respondent Group Comparisons

Youth were significantly more likely than parents to endorse positive MC beliefs (“cannabis is a safe and effective treatment for pediatric mental health conditions”) (see Table 2 and Table 3). They also had more favorable attitudes about youth cannabis, MC, and CBD use; lower cannabis and MC risk perceptions; and higher perceptions of cannabis, MC, and CBD health benefits compared to parents. No group differences were observed for CBD-related medical beliefs, risk perceptions, or MEEQ-B expectancy scores. Results were unchanged when analyses were rerun, covarying for total psychiatric problems.

We conducted exploratory analyses (Appendix A) to determine if attitudinal differences varied as a function of use patterns or product types. Results were consistent across every frequency (trying once or twice, occasional use, and regular use), method of administration (smoking, vaping, oral [edibles], and concentrates), and chemotype (high THC, low THC, and high CBD).

### 3.4. State-Level CL and Group-by-State CL Comparisons

Cannabis- and CBD-related medical beliefs, attitudes, and health benefit perceptions in the total sample varied as a function of state CL status (Table 2, Table 3 and Appendix A). In post hoc analyses, respondents living in MCL states (vs. No-CL states) were more likely to report positive or neutral medical beliefs about cannabis and were more likely than RCL and No-CL respondents to report neutral CBD medical beliefs. RCL respondents had higher health benefit perceptions about youth MC use and more positive attitudes about youth CBD use compared to MCL and No-CL respondents, respectively. Additionally, a significant group-by-state CL interaction effect was observed. In post hoc analyses, No-CL (vs. RCL) parents were more uncertain about the risk for harm related to youth MC and CBD use.

In post hoc state CL analyses stratified by respondent group, RCL vs. MCL youth were less likely to report unknown risk perceptions and more likely to report moderate/great health benefit perceptions for regular youth cannabis use. RCL vs. No-CL parents were also less likely to report unknown risk perception for regular cannabis use. MCL vs. No-CL parents were more likely to report positive medical beliefs about cannabis, while RCL vs. MCL parents had more approving attitudes towards youth MC and CBD.

### 3.5. Factors Associated with Youth Intentions to Use Cannabis and CBD

We could not examine factors associated with intention to use CBD due to insufficient variability in responses. Regarding cannabis use intentions, exploratory ordinal regression models (Appendix A) revealed that mean age; presence of lifetime psychiatric symptoms of impulsivity, trauma reactivity, eating disorders, non-suicidal self-injurious behaviors, suicidal thoughts and behaviors; total psychiatric problem scores; MEEQ-B positive expectancy scores; attitudes (approving and neutral > disapproving), risk perceptions (great/moderate risk < no/low risk), and health benefit perceptions (great/moderate benefit > no/low benefit) related to youth cannabis use were associated with intentions to use cannabis among youth after controlling for age, sex, parent education, and site. We identified no significant associations in ordinal regression models between cannabis use intentions and sex, medical beliefs about cannabis, GAD-2, PHQ-2, or MEEQ-B negative expectancy scores, past year CBD use by youth or a household member, and past year household member MC use. Youth cannabis use intentions also did not vary as a function of state CL status.

Lifetime mental health symptom categories correlated with cannabis use intentions were highly collinear with each other and total psychiatric problem scores. We reran psychiatric symptom category models, controlling for all previous covariates and total psychiatric problem scores. Depression (β (95% CI) = 1.33 (0.11, 2.56), *p* = 0.03), sleep problems (β (95% CI) = −0.91 (−1.74, −0.08), *p* = 0.03), and alcohol or drug use problems (β (95% CI) = 1.61 (0.51, 2.72), *p* = 0.004) were significant predictors of cannabis use intentions. We then excluded specific psychiatric symptom categories but retained the total psychiatric problem score in the final multivariable model which is presented in Table 4. Male sex (*p* = 0.01), MEEQ-B positive expectancy scores (*p* = 0.007), and positive attitudes about youth cannabis use (*p* < 0.001) were associated with greater cannabis use intention among youth. Associations between total psychiatric problem scores and cannabis use intention in youth approached significance (*p* = 0.08); cannabis use intention associations with mean age, perceived risk for harm, and perceived health benefit related to youth cannabis use were not significant.

### 3.6. Sensitivity Analyses

Associations between cannabis and CBD-related medical beliefs, attitudes, perceptions, and expectancies that differed as a function of respondent group or state CL status were robust to sensitivity testing. Effects were largely consistent between the final model and various sensitivity analyses (see Appendix A).

## 4. Discussion

To our knowledge, this is the first study to examine attitudes, beliefs, and behaviors for cannabis, MC, and CBD use in youth with psychiatric disorders, and one of only a few studies to compare youth and parent attitudes and behaviors and to investigate differences in attitudes and behaviors in youth and parents living in states with differing CL. This study provides an initial characterization of multiple factors relevant to cannabis prevention efforts.

Results from this study show that US youth receiving mood disorder treatment and their parents perceive cannabis and CBD products to be safe and effective treatments for multiple pediatric mental health conditions. Most youth and parents in the sample reported believing that regular use of cannabis and CBD products reduces depression and anxiety, with nearly half of the youth and one-third of the parent respondents believing cannabis use to be beneficial for suicidal thoughts and behaviors, in direct contrast to the current evidence [11,12]. Additionally, one in five youth respondents reported believing that cannabis is beneficial in treating psychosis, which is concerning in light of research showing links between cannabis use and psychosis [23]. Our results are consistent with previous studies showing trends towards more positive cannabis-related attitudes and medical beliefs, elevated perception of health benefits, reduced perception of risk for harm, and increased use rates across the entire US adult population and subgroups of US adolescents and young adults [32,34,35,36,38]. Additionally, our youth findings are consistent with prior studies on CBD attitudes in young adults [50], youth who use cannabis [51], and individuals with substance use disorders [52]. Our parent findings align with a recent study by Schwaller and colleagues that found positive safety perceptions about unlicensed CBD products and high administration rates of these products to children with ADHD, autism spectrum disorder, and anxiety disorder by US parents who had a 7–18-year-old child diagnosed with those conditions [53]. Our attitudinal results also parallel findings from international studies, which have shown shifting societal attitudes, policies, and use patterns related to both cannabis and CBD in many countries around the world [25,54,55,56,57]. Our study findings build on national survey studies showing that trends towards more favorable attitudes and perceptions about cannabinoids are also occurring in vulnerable subgroups, in this case, youth with mood disorders [25,26,27]. These findings are of particular importance, given the current mental health crisis in US youth and the US mental healthcare system’s struggle to respond to this crisis, made more complicated by the rapid growth of multibillion-dollar cannabis and CBD product markets, which have promoted broad claims of health benefits unsupported by the scientific literature [6,58]. Recent meta-analyses and longitudinal studies show that cannabis use in adolescents with and without mood disorders is associated with poorer mood outcomes, reduced treatment response in those receiving mental healthcare, and increased risk for depression, suicidal ideations, and suicide attempts by young adulthood [11,12]. Concerningly, regular cannabis use and cannabis use disorders may increase mortality in young people with mood disorders. A recent study by Fontanella et al. (2021) that examined Ohio Medicaid Claims data from 204,780 US youths (ages 10–24) found that, among youth with mood disorders, having a cannabis use disorder diagnosis was associated with increased non-fatal self-harm and all-cause mortality [17]. In contrast to the evidence for harm from cannabis use, the evidence for benefit from cannabinoid-based therapeutics for psychiatric conditions remains in its infancy [59,60,61]. Safety and efficacy data on CBD (and other cannabinoids) as therapeutic agents for mood disorders remain limited, especially in youth samples; although early studies have shown promise for certain conditions (e.g., anxiety), and multiple clinical trials are underway [5,59,61,62]. A recent systematic review by Hasbi et al. (2023) examined the therapeutic role of cannabinoids, including THC and CBD, for psychiatric conditions and concluded that the evidence that cannabinoids improve depression or anxiety disorders is currently weak and of low quality [63]. In sum, there is currently moderate to strong scientific evidence showing elevated risk for negative health outcomes from cannabis exposure in youth with and without mood disorders and insufficient evidence for the safety and efficacy of cannabis or CBD products for pediatric neuropsychiatric conditions beyond rare treatment-resistant seizure disorders [5,60]. Taken within this context, our attitudinal findings point to low cannabinoid health literacy and significant discrepancies between what youth with mood disorders and their parents believe about the risks and benefits of cannabis and CBD use and the current evidence related to safety and efficacy of these products, particularly in relation to cannabis’s effects on mental health and the therapeutic use of CBD and other cannabinoids for the treatment of mood disorders, psychotic disorders, and suicidal behaviors.

We also found evidence of intergenerational differences in cannabis and CBD-related beliefs, attitudes, and perceptions between youth and parent groups that remained significant after controlling for different confounding variables and were robust to sensitivity tests. Consistent with prior studies, in general, and population samples, our results showed that youth with mood disorders were significantly more likely to report believing that cannabis is safe and effective for pediatric mental health conditions and to have a lower perception of risk for harm related to youth cannabis use, and a greater perception of health benefit related to youth cannabis and CBD use than parents [40]. Intergenerational differences in attitudes, beliefs, and perceptions seen in this study were observed across all cannabinoid product types, chemotypes, THC/CBD potencies, use frequencies, and methods of administration. This suggests a possible class effect for cannabinoids by age/generational group. Group differences in perceptions and attitudes could also be explained by developmental differences in sensation seeking, impulsiveness, and reward and risk processing between youth and adult respondents [64].

One of the main objectives of this study was to examine the influence of state-level CL on cannabis and CBD attitudes, expectancies, health perceptions, and behaviors in youth and parent respondents. Cannabis and CBD-related medical beliefs, attitudes about medical cannabis and CBD, and health benefit and risk perceptions about cannabis and MC use varied as a function of state-level CL, with some differences between youth and parent respondents. Interestingly, our results exhibited some evidence of a gradient effect whereby respondents in RCL states had more favorable attitudes than respondents in MCL states, who generally had more favorable attitudes than respondents in No-CL states. Notably, parents living in RCL states had more approving attitudes of youth MC and CBD use, and youth living in RCL states had greater health benefit perceptions about MC. Our CL findings are consistent with previous studies showing an impact of CL on cannabis-related attitudes, medical beliefs, perceptions, and use behaviors in adolescents and adults [14,39,65,66,67,68], although null results have also been found [69], suggesting heterogeneous associations with CL impacting some attitudes and beliefs more so than others [38]. Our study also extends this research to include CBD-related attitudes and medical beliefs in this population, with our results showing that these attitudes and beliefs also differed as a function of state CL status. Variation between our results and those found in other studies may be related to differences in population characteristics, recruitment/sampling frame, sample size, study design, covariate selection, and analytic strategy. Our findings showing more favorable cannabis and CBD attitudes among youth with mood disorders and parents who live in RCL states may have public health relevance. At the same time, it is important to interpret these findings cautiously, given the sample size and cross-sectional design of the study, which preclude causal or directional determination. Future research is needed to replicate this work and further clarify the impact of changing cannabis policies on attitudes, perceptions, and behaviors in vulnerable populations, and should be used to guide the development of targeted cannabis prevention and educational messaging for these groups.

Our results also identified sociodemographic, clinical, and attitudinal factors associated with intentions to use cannabis among youth with mood disorders. Consistent with previous studies [43,70,71,72,73,74], factors associated with cannabis use intentions among youth in our sample included older age, sex, total psychiatric problem scores, lifetime presence of certain psychiatric symptom categories (impulsivity, post-traumatic stress disorder, eating disorders, non-suicidal self-injurious behavior, and suicidal thoughts and/or behaviors), positive cannabis use expectancies, favorable cannabis attitudes, and lower risk and higher health benefit perceptions related to cannabis use. Among these, male sex, positive cannabis expectancies, and favorable cannabis attitudes were independently associated with cannabis use intentions after adjusting for age, sex, parent education, site, and each of the other variables in our multivariate analysis.

Based on our study findings, goals of next-stage multilevel prevention efforts should focus on cannabinoid health literacy, targeting youth and parent misperceptions, providing education about mental health risks related to cannabis use, while concurrently promoting aggressive treatment of youth mood disorders to reduce the likelihood these youth will initiate cannabis or progress to regular use. Attitudinal and mental health factors represent important and malleable intervention targets and can inform public health cannabis messaging efforts [58,73], especially as youth living in RCL states have more favorable cannabis attitudes associated with increased intention to use. Cannabis prevention efforts for this population should largely be driven by pediatric medical and mental health clinicians; public health campaigns should focus on cannabis health literacy by providing targeted, evidence-based education to youth and parents and encouraging fact-driven discussions between parents, youth, and providers about cannabinoids and mental health. Standardized medical/healthcare education and workforce training for pediatric mental and medical healthcare providers about cannabinoid risks and benefits, and how to screen for cannabinoid product use and advise youth/families, are needed. In order to be effective, this medical/healthcare education on cannabis and cannabinoids should be included as required training content for residency/fellowship programs and continued medical education content for board certification and maintenance for psychiatrists, psychologists, pediatricians, primary care providers, and other licensed mental health clinicians. Importantly, taking lessons from the history of the alcohol and tobacco industry’s influence on medical/healthcare education, research, and practice [75,76], medical education content on cannabis and cannabinoids should be unbiased and evidence-informed, and should consider and weigh the risks and potential benefits from different cannabinoid products. This educational content should be prepared and disseminated by experts in cannabis prevention and early intervention research without cannabis industry influence. Furthermore, future cannabis regulatory and policy reforms should incorporate strong evidence-informed risk-mitigating provisions and resources that seek to reduce cannabis use in youth and protect vulnerable populations [20].

This study has several relevant limitations and some notable strengths. Given self-report data, recall or social desirability bias might have influenced response choices and led to over- or under-reporting of desirable or undesirable attitudes or behaviors. One study limitation is related to measurement validation and content. While we used a formative content validation approach to develop the survey and included items from validated instruments, the full survey has not been psychometrically validated. Notably, some content domains explored in this report related to cannabis outcome perceptions did not have validated instruments at the time of data collection. Development and psychometric validation of instruments that measure use behaviors, attitudes, beliefs, perceptions, and expectancies for different cannabinoid-based products is an important area of research focus for the field that is needed to improve measurement validity for future studies [77]. Regarding missing content, we did not collect detailed information on patterns of non-medical/recreational cannabis use, limiting our ability to examine associations between cannabis use and attitudinal factors, although we did collect data on youth alcohol/drug use problems and controlled for this in sensitivity analyses. We also did not collect information about medical comorbidities in youth, and, thus, were unable to control for potential confounding effects from comorbid medical conditions (e.g., chronic pain, cancer, and seizure disorders) that MC and CBD are sometimes used to treat. The absence of race and ethnicity data is another notable limitation of this study, especially given evidence for racial and ethnic differences in cannabis use [78,79] and negative health outcomes related to cannabis use [80], along with racial/ethnic disparities in cannabis use disorder treatment access and cannabis-related arrests that disproportionately impact Black and Indigenous people of color [80,81]. Future research should test for racial and ethnic differences and examine the role of minority stress on cannabis-related attitudes, use intentions, and behaviors. Another limitation of the study is related to sample size and generalizability. Our sample size was relatively small, particularly when stratifying by group and state-level CL status, resulting in some underpowered analyses. It is important to note that the sample is a convenience sample. All youth respondents were receiving mental health treatment in child mood clinics linked to US academic medical centers. Cannabis and CBD-related attitudes, beliefs, perceptions, and behaviors from this population may not generalize to non-treatment-engaged youth with mood disorders or those receiving treatment in different settings/contexts. Most parents were mothers; fathers and non-traditional caregivers were underrepresented. Future studies should seek to draw larger samples with greater diversity, obtain an equal number of respondents from RCL, MCL, and No-CL states, and consider oversampling underrepresented or vulnerable subgroups. The study also has relevant strengths. We comprehensively characterized individual-level, treatment-level, and environmental factors associated with youth cannabis use intentions. We focused on youth with mood disorders, a vulnerable subgroup that experiences poorer health outcomes related to cannabis exposure. Broad geographic recruitment and respondents’ variability in lived experience with No-CL, MCL, and RCL conditions allowed us to test the effects of state-level CL on cannabis and CBD-related attitudes, beliefs, and perceptions.

## 5. Conclusions

This study found that US youth receiving mood disorder treatment and their parents perceive cannabis and CBD to be safe and effective for mental health problems and to have a low risk for harm from regular use. Further, it identified factors that increase the likelihood of future cannabis use in youth. These findings provide a roadmap to inform next-stage cannabis prevention intervention and public health messaging efforts for youth receiving mental health treatment in this era of changing CL.

## Figures and Tables

**Table 1 ijerph-22-01576-t001:** Sample Characteristics of MABS Survey Participants by Respondent Group Status.

Variable	Respondent Group		Youth vs. Parent	
	Youth (*n* = 84)	Parent (*n* = 66)	Statistic	*p*-Value
Age, years, Mean (SD)	17.55 (3.30)	47.36 (6.14)	*t* = −38.60	<0.001
Sex, *n* (%)			χ^2^ = −1.59	0.12
Female	55 (65.5%)	52 (78.8%)		
Male	27 (32.1%)	14 (21.2%)		
Non-binary	1 (1.2%)	0 (0.0%)		
Parent education, *n* (%)			*t =* −2.63	0.01
Some high school	15 (17.9%)	3 (4.5%)		
Completed high school	7 (8.3%)	4 (6.1%)		
Some college or university	16 (19.0%)	14 (21.2%)		
Completed college or university	44 (52.4%)	45 (68.2%)		
Respondents living in states, by state-level CL status, *n* (%)			*t =* 0.80	0.42
Recreational cannabis laws	35 (41.7%)	28 (42.2%)		
Medical cannabis laws	33 (39.3%)	18 (27.3%)		
No cannabis laws	16 (19.0%)	20 (30.3%)		
Parent-reported relationship with youth receiving MH tx, *n* (%)			NA	NA
Mother	--	48 (72.7%)		
Father	--	11 (16.6%)		
Adoptive mother	--	5 (7.6%)		
Adoptive father	--	2 (3.0%)		
Other	--	0 (0.0%)		
Parent-reported age of youth receiving MH tx, yrs, Mean (SD)		16.02 (2.45)	NA	NA
Parent-reported sex of youth receiving MH tx, *n* (%)			NA	NA
Female	--	37 (56.9%)		
Male	--	28 (43.1%)		
Non-binary	--	0 (0.0%)		
Lifetime psychiatric symptom categories/conditions in youth, *n* (%)				
Depression	68 (81.0%)	51 (77.3%)	χ^2^ = 0.55	0.58
Anxiety/worry	75 (89.3%)	59 (89.4%)	χ^2^ = −0.02	0.98
Anger/irritability	24 (28.6%)	18 (27.3%)	χ^2^ = 0.18	0.96
Sleep problems	38 (45.2%)	20 (30.3%)	χ^2^ = 1.87	0.06
Trauma reactivity	22 (26.2%)	11 (16.7%)	χ^2^ = 1.40	0.16
Alcohol or drug use problems	9 (10.7%)	9 (13.6%)	χ^2^ = −0.54	0.59
Disordered eating behaviors	21 (25.0%)	5 (7.6%)	χ^2^ = 2.86	0.01
Bipolar disorder or mania	17 (20.2%)	4 (6.1%)	χ^2^ = 2.52	0.01
Psychosis, delusions, hallucinations	10 (11.9%)	4 (6.1%)	χ^2^ = 1.22	0.23
Non-suicidal self-injurious thoughts and behaviors	36 (42.9%)	13 (19.7%)	χ^2^ = 3.08	0.01
Suicidal thoughts and behaviors	38 (45.2%)	19 (28.8%)	χ^2^ = 2.08	0.04
Total psychiatric problem score ^A^ Mean (SD)	5.74 (3.49)	4.53 (2.60)	*t =* 2.35	0.02
Current MH treatment type, *n* (%)				
Individual therapy	72 (85.7%)	63 (95.5%)	χ^2^ = −1.98	0.04
Group therapy	7 (8.3%)	19 (28.8%)	χ^2^ = −3.39	<0.001
Psychiatric medication	59 (70.2%)	49 (74.2%)	χ^2^ = −0.54	0.59
PHQ-2 depression total score, Mean (SD)	2.33 (1.86)	--	NA	NA
Youth with current depressive disorder ^B^ *n* (%)	33 (39.3%)	--	NA	NA
GAD-2 anxiety total score, Mean (SD)	2.76 (1.98)	--	NA	NA
Youth with current anxiety disorder ^C^ *n* (%)	39 (46.4%)	--	NA	NA

Note: Demographic and clinical characteristics of the MABS sample are presented in this table for the entire sample, stratified by respondent group status (youth vs. parent). Youth self-report and parent-report MABS surveys differed slightly in their composition with regard to demographic questions. When responses from both respondent groups are shown, results of statistical tests comparing groups on these variables are also presented. Group comparisons were made using independent *t*-tests and chi-squares for continuous and categorical variables, respectively. It is important to note that, while both youth and parent respondents were recruited from the same sites, this sample does not represent youth–parent dyads. Thus, different responses to overlapping questions are expected. Abbreviations: NA = Not applicable; MH tx = Mental health treatment; SD = standard deviation; Patient Health Questionnaire—2-item version = PHQ-2; Generalized Anxiety Disorder Questionnaire—2-item. ^A^ Total psychiatric problem score: This is a diagnostic variable we created to index total psychiatric problems for each participant, calculated by coding each category as 1 = present and 0 = absent and summing across categories. ^B^ Current depressive disorder diagnosis based upon PHQ-2 cutoff ≥ 3 [46]. ^C^ Current anxiety disorder diagnosis based upon GAD-2 cutoff ≥ 3 [48].

**Table 2 ijerph-22-01576-t002:** Multinominal and Linear Regression Models Examining Cannabis-Related Attitudes, Perceptions, and Expectancies, by Respondent Group and State-Level Cannabis Law Status, Covarying for Age, Sex, Parent Education, and Site.

		Respondent Group		Statistics	State-Level Cannabis Law Status			Statistics		
	Full Sample: All Respondents (*n* = 150)	Youth Respondents (*n* = 84)	Parent Respondents (*n* = 66)	Youth vs. Parents	Respondents from States Without CL (*n* = 36)	Respondents from States with MCL (*n* = 51)	Respondents from States with RCL (*n* = 63)	MCL vs. No-CL	No-CL vs. RCL	MCL vs. RCL
MC is a safe and effective treatment										
Agree	103 (72.0%)	61 (78.2%)	42 (64.6%)	3.98 (1.34–11.78) *	24 (68.6%)	36 (72.0%)	45 (71.4%)	3.91 (1.15–13.37) *	0.52 (0.17–1.56)	1.63 (0.44–5.98)
Neither/nor	18 (12.6%)	11 (14.1%)	7 (10.8%)	3.46 (0.83–14.34)±	2 (5.7%)	10 (20.0%)	9 (14.3%)	12.94 (1.93–86.69) **	0.17 (0.03–1.11)±	1.93 (0.41–9.20)
Disagree	22 (15.4%)	6 (7.7%)	16 (24.6%)	Ref.	9 (25.7%)	4 (8.0%)	9 (14.3%)	Ref.	Ref.	Ref.
Attitude about regular cannabis use by youth										
Approve	25 (16.7%)	24 (28.6%)	1 (1.5%)	32.50 (4.12–256.38) ***	5 (13.9%)	9 (18.4%)	11 (17.5%)	0.83 (0.23–2.96)	0.68 (0.19–2.42)	0.87 (0.29–2.65)
Neither/nor	22 (14.7%)	18 (21.4%)	4 (6.1%)	8.12 (2.41–27.40) ***	2 (5.6%)	9 (18.4%)	11 (17.5%)	2.22 (0.43–11.47)	0.34 (0.06–1.79)	1.10 (0.36–3.33)
Disapprove	103 (68.7%)	42 (50.0%)	61 (92.4%)	Ref.	29 (80.6%)	31 (63.3%)	41 (65.1%)	Ref.	Ref.	Ref.
Attitude about MC use by youth										
Approve	72 (50.3%)	54 (69.2%)	18 (27.7%)	11.62 (4.42–30.54) ***	13 (36.1%)	28 (54.9%)	33 (52.2%)	1.99 (0.72–5.56)	0.37 (0.12–1.11)±	0.54 (0.19–1.51)
Neither/nor	29 (20.3%)	16 (20.5%)	13 (20.0%)	4.50 (1.48–13.67) **	9 (25.0%)	7 (13.7%)	15 (23.8%)	1.03 (0.32–3.33)	0.56 (0.17–1.87)	0.36 (0.10–1.29)
Disapprove	42 (29.4%)	8 (10.3%)	34 (52.3%)	Ref.	14 (38.9%)	16 (31.4%)	15 (23.8%)	Ref.	Ref.	Ref.
Perceived risk for harm related to regular cannabis use by youth										
Mod/high risk	106 (70.7%)	52 (61.9%)	54 (81.8%)	0.41 (0.18–0.96) *	26 (72.5%)	36 (70.6%)	44 (69.8%)	1.13 (0.42–3.07)	1.20 (0.44–3.31)	1.24 (0.49–3.10)
Do not know	7 (4.7%)	5 (6.0%)	2 (3.0%)	0.69 (0.10–4.65)	2 (5.6%)	3 (5.9%)	2 (3.2%)	0.97 (0.14–6.76)	1.77 (0.20–15.82)	1.50 (0.20–11.11)
No/low risk	37 (24.7%)	27 (32.1%)	10 (15.2%)	Ref.	8 (22.2%)	12 (23.5%)	17 (27.0%)	Ref.	Ref.	Ref.
Perceived risk for harm related to MC use by youth										
Mod/high risk	43 (30.1%)	14 (17.9%)	29 (44.6%)	0.25 (0.11–0.56) ***	13 (36.1%)	13 (25.5%)	19 (30.2%)	0.64 (0.25–1.63)	1.52 (0.59–3.96)	1.00 (0.40–2.50)
Do not know	12 (8.4%)	7 (9.0%)	5 (7.7%)	0.58 (0.15–2.17)	4 (11.1%)	7 (13.7%)	2 (3.2%)	0.85 (0.22–3.25)	3.69 (0.61–22.42)	3.79 (0.71–20.25)
No/low risk	88 (61.5%)	57 (73.1%)	31 (47.7%)	Ref.	19 (52.8%)	31 (60.8%)	42 (66.7%)	Ref.	Ref.	Ref.
Perceived benefit for mental health related to regular cannabis use by youth										
Mod/high benefit	39 (26.0%)	32 (38.1%)	7 (10.6%)	5.69 (2.23–14.56) ***	9 (25.0%)	12 (23.5%)	18 (28.6%)	0.90 (0.32–2.52)	0.70 (0.25–2.02)	0.62 (0.23–1.66)
Do not know	10 (6.7%)	10 (11.9%)	0 (0.0%)	--	0 (0.0%)	6 (11.8%)	4 (6.3%)	--	--	1.37 (0.30–6.17)
No/low benefit	101 (67.3%)	42 (50.0%)	59 (89.4%)	Ref.	27 (75.0%)	33 (64.7%)	41 (65.1%)	Ref.	Ref.	Ref.
Perceived benefit for mental health related to MC use by youth										
Mod/high benefit	62 (43.4%)	44 (56.4%)	18 (27.7%)	4.76 (2.21–10.25) ***	15 (41.7%)	16 (31.4%)	34 (54.0%)	1.02 (0.42–2.50)	0.52 (0.20–1.32)	0.33 (0.13–0.83) *
Do not know	12 (8.4%)	12 (15.4%)	0 (0.0%)	--	1 (2.8%)	8 (15.7%)	4 (6.3%)	6.28 (0.65–60.69)	0.28 (0.03–3.07)	1.45 (0.31–6.82)
No/low benefit	69 (48.3%)	22 (28.2%)	47 (72.3%)	Ref.	20 (55.6%)	27 (52.9%)	25 (39.7%)	Ref.	Ref.	Ref.
Cannabis Effect Expectancies										
MEEQ-B positive exp.	3.74 (0.75)	3.74 (0.81)	3.69 (0.66)	−0.07 (−0.36–0.15)	3.61 (0.86)	3.79 (0.62)	3.72 (0.77)	0.09 (−0.19, 0.45)	0.05 (−0.26–0.43)	−0.05 (−0.36–0.22)
MEEQ-B negative exp.	3.43 [0.77]	3.32 (0.75)	3.54 (0.76)	0.11 (−0.08–0.42)	3.40 (0.91)	3.41 (0.68)	3.43 (0.73)	0.00 (−0.34, 0.36)	0.04 (−0.26–0.39)	0.03 (−0.23–0.32)

Note: Multinomial logistic regression models included cannabis- and medical-cannabis-related medical beliefs; attitudes; risk perceptions; health benefit perceptions as outcome variables (reference group for each variable shown above) with respondent group membership as a between-subjects factor covarying for sex, parent education; the site for respondent group analyses and with state-level CL status as between subjects factor covarying for age, sex, and parent education for CL analyses. For MC beliefs, attitudes, and perceptions, the analytic sample included 143 resondents (95.3% of the total sample) who reported that they had heard of MC and completed MC-related items from the MABS survey instrument. Linear regression models included MEEQ-B positive and negative expectancy scores as outcome variables with respondent group membership as a between-subjects factor covarying for sex, parent education, as well as the site for respondent group analyses and with state-level CL status as a between-subjects factor covarying for age, sex, and parent education for CL analyses. Results from the multinomial logistic regressions are presented in the form of adjusted odds ratios (AORs) with 95% confidence intervals (CIs). Results from linear regression models are presented in the form of standardized coefficient beta/difference scores with 95% CI. Statistical significance is shown with ± for *p* < 0.10, * for *p* < 0.05, ** for *p* < 0.01, and *** for *p* < 0.001. Some comparisons had dependent variable levels in one of the comparator subpopulations with zero frequencies, making statistical comparison incalculable (demarcated with --). Abbreviations: No-CLs = No cannabis laws; MC = medical cannabis; MCLs = MC Laws; RCL = Recreational Cannabis Law; MEEQ-B pos exp = Marijuana Effect Expectancy Questionnaire—Brief Version positive expectancy score; MEEQ-B negative exp = MEEQ-B negative expectancy score.

**Table 3 ijerph-22-01576-t003:** Multinominal and Linear Regression Models Examining Cannabidiol-Related Attitudes, Perceptions, and Expectancies, by Respondent Group and State-Level Cannabis Law Status, Covarying for Age, Sex, Parent Education, and Site.

		Respondent Group		Statistics	State-Level Cannabis Law Status			Statistics		
	Full Sample: All Respondents (*n* = 150)	Youth Respondents (*n* = 84)	Parent Respondents (*n* = 66)	Youth vs. Parents	Respondents from States Without CL (*n* = 36)	Respondents from States with MCL (*n* = 51)	Respondents from States with RCL (*n* = 63)	MCL vs. No-CL	No-CL vs. RCL	MCL vs. RCL
CBD is a safe and effective treatment										
Agree	99 (71.7%)	55 (74.3%)	44 (68.8%)	2.19 (0.64–7.48)	24 (72.7%)	31 (64.6%)	44 (77.2%)	3.05 (0.66–14.08)	0.64 (0.17–2.37)	1.27 (0.28–5.78)
Neither/nor	25 (18.1%)	14 (18.9%)	11 (17.2%)	2.36 (0.56–9.87)	4 (12.1%)	14 (29.2%)	7 (12.3%)	8.13 (1.28–51.37) *	0.64 (0.11–3.67)	4.43 (0.77–25.49) ^±^
Disagree	14 (10.1%)	5 (6.8%)	9 (14.1%)	Ref.	5 (15.2%)	3 (6.3%)	6 (10.5%)	Ref.	Ref.	Ref.
Attitude about CBD use by youth										
Approve	74 (53.6%)	48 (64.9%)	26 (40.6%)	5.71 (2.07–15.78) ***	14 (42.4%)	26 (54.2%)	33 (58.9%)	2.10 (0.70–6.29)	0.26 (0.07–0.88) *	0.32 (0.09–1.07) ^±^
Neither/nor	35 (25.5%)	19 (25.7%)	16 (25.0%)	3.42 (1.11–10.50) *	9 (27.3%)	11 (22.9%)	15 (26.8%)	1.46 (0.43–4.91)	0.38 (0.10–1.43)	0.32 (0.08–1.25)
Disapprove	29 (21.0%)	7 (9.5%)	22 (34.4%)	Ref.	10 (30.3%)	11 (22.9%)	8 (14.3%)	Ref.	Ref.	Ref.
Perceived risk for harm related to CBD use by youth										
Mod/high risk	29 (21.0%)	12 (16.2%)	17 (26.6%)	0.54 (0.22–1.33)	8 (24.2%)	11 (22.9%)	10 (17.5%)	0.88 (0.31–2.51)	1.93 (0.62–6.06)	1.78 (0.62–5.12)
Do not know	11 (8.0%)	6 (8.1%)	5 (7.8%)	0.80 (0.21–3.02)	4 (12.1%)	6 (12.5%)	1 (1.8%)	0.85 (0.22–3.33)	7.95 (0.83–76.55) ^±^	8.45 (0.91–78.82) ^±^
No/low risk	98 (71.0%)	56 (75.7%)	42 (65.6%)	Ref.	21 (63.6%)	31 (64.6%)	46 (80.7%)	Ref.	Ref.	Ref.
Perceived benefit for mental health related to CBD use by youth										
Mod/high benefit	59 (42.8%)	38 (51.4%)	21 (32.8%)	2.69 (1.26–5.74) *	12 (36.4%)	18 (37.5%)	29 (50.9%)	1.41 (0.56–3.51)	0.46 (0.18–1.20)	0.52 (0.21–1.30)
Do not know	12 (8.7%)	11 (14.9%)	1 (1.6%)	19.45 (2.33–162.32) ***	1 (3.0%)	7 (14.6%)	4 (7.0%)	6.44 (0.70–59.10) ^±^	0.26 (0.03–2.68)	1.42 (0.33–6.15)
No/low benefit	67 (48.6%)	25 (33.8%)	42 (65.6%)	Ref.	20 (60.6%)	23 (47.8%)	24 (42.1%)	Ref.	Ref.	Ref.

Note: Multinomial logistic regression models included CBD-related medical beliefs; attitudes; risk perceptions; health benefit perceptions as outcome variables (reference group for each variable shown above) with respondent group membership as a between-subjects factor covarying for sex, parent education; the site for respondent group analyses; state-level CL status as a between-subjects factor covarying for age, sex, and parent education for CL analyses. For CBD beliefs, attitudes, and perceptions, the analytic sample included 138 resondents (92% of the total sample) who reported that they had heard of CBD and completed CBD-related items from the MABS survey instrument. Results from the multinomial logistic regressions are presented in the form of adjusted odds ratios (AORs) with 95% confidence intervals (CIs). Results from linear regression models are presented in the form of standardized coefficient beta/difference scores with 95% CI. Statistical significance is shown with ± for *p* < 0.10, * for *p* < 0.05, ** for *p* < 0.01, and *** for *p* < 0.001. Some comparisons had dependent variable levels in one of the comparator subpopulations with zero frequencies, making statistical comparison incalculable (demarcated with --). Abbreviations: CBD = cannabidiol; No-CLs = No cannabis laws; MC = medical cannabis; MCLs = MC Laws; RCLs = Recreational Cannabis Laws.

**Table 4 ijerph-22-01576-t004:** Multivariable Ordinal Regression Model Showing Factors Associated with Intentions to Use Cannabis Among US Youth Receiving Treatment For Mood Disorders.

Factor	Grouping Variable: Intention to Use Cannabis				
	Participants Responding “Definitely” or “Probably” (*n* = 23)	Participants Responding “Possibly” (*n* = 7)	Participants Responding “Definitely Not” or “Probably Not” (*n* = 54)	Parameter Estimate (95% CI)	*p*-Value
Mean Age (SD)	18.39 (2.89)	19.86 (3.18)	16.89 (3.31)	0.02 (−0.01, 0.05)	0.26
Sex, *n* (%)					
Male	9 (39.1%)	0 (0.0%)	18 (34.0%)	1.08 (0.20, 1.97)	0.017
Female	14 (60.9%)	7 (100.0%)	35 (66.0%)	Ref.	
Total psychiatric problem score	7.22 [3.25]	6.29 [3.04]	5.04 [3.48]	0.11 (−0.01, 0.24)	0.078
MEEQ-B positive exp. score	4.29 [0.55]	3.95 [0.36]	3.48 [0.83]	0.90 (0.25, 1.56)	0.007
Attitude about regular cannabis use by youth					
Approve	11 (47.8%)	2 (28.6%)	11 (20.4%)	1.02 (−0.28, 2.33)	0.12
Neither/nor	7 (30.4%)	4 (57.1%)	7 (13.0%)	2.01 (0.88, 3.15)	<0.001
Disapprove	5 (21.7%)	1 (14.3%)	36 (66.7%)	Ref.	
Perceived risk related to youth regular cannabis use					
Mod/high risk	10 (43.5%)	4 (57.1%)	38 (70.4%)	−0.34 (−1.34, 0.65)	0.50
Do not know	0 (0.0%)	0 (0.0%)	5 (9.3%)	0.32 (−2.16, 2.80)	0.80
No/low risk	13 (56.5%)	3 (42.9%)	11 (20.4%)	Ref.	
Perceived health benefits related to youth regular cannabis use					
Mod/high benefit	15 (65.2%)	0 (0.0%)	17 (31.5%)	−0.15 (−1.13, 0.83)	0.76
Do not know	1 (4.3%)	1 (14.3%)	8 (14.8%)	−0.66 (−2.63, 1.32)	0.51
No/low benefit	7 (30.4%)	6 (85.7%)	29 (53.7%)	Ref.	

Note: Multivariable logistic regression models included the intention to use cannabis as the outcome variable, with all variables in this table as simultaneous regressors. Factors included as independent variables in the model were selected based on being significantly associated with intention to use cannabis in exploratory ordinal regression models, adjusting for age, sex, parent education, and site (see Appendix A). Sex was also included as a covariate in the model. Results are presented as standardized beta parameter estimates with 95% confidence intervals (CI). Abbreviation: MEEQ-B positive exp = Marijuana Effect Expectancy Questionnaire—Brief Version positive expectancy score; SD = standard deviation.

## Data Availability

Data is unavailable due to privacy concerns or ethical restrictions.

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
