# Peer review of "Differences in Cannabis and Cannabidiol Attitudes, Perceptions, and Behaviors Between US Adolescents Receiving Mood Disorder Treatment and Their Parents Across Legal Contexts"

_ijerph, 2025, doi:10.3390/ijerph22101576_

Round 1

Reviewer 1 Report

Comments and Suggestions for Authors

This well-crafted article addresses an important issue for adolescents and young adults, given the high rates of use of cannabis by these populations and the lack of knowledge about its potential negative effects. This article focuses on a subpopulation of adolescents and young adults with mood disorders.

Comments for specific sections:

Abstract - concise, includes pertinent findings

Introduction - Describes well what is known about attitudes and expectancies regarding cannabis use and what is known about the role of legalization of either recreational or medical marijuana. Outlines the current aims clearly.

Measures -

The authors utilize a survey on marijuana attitudes and beliefs that they have created de-novo. This reviewer would have preferred more information about this survey in the body of the paper, not as a supplement, to include number of items, domain of items, etc. Also, there was no attempt to provide even preliminary validation of this instrument (which is commented upon in the Discussion). However, basic psychometric analysis could have been done to provide some sense of its properties and validity. 

Statistical Analyses - Discussion is relevant and analytic plan seems robust.

Results - Clearly stated and readily understood.

Discussion - Overall, the discussion addresses the results directly and is clearly written. Strengths and limitations are addressed. Recognizing the word-limit of articles, this author still would have liked even a limited mention of the some of the data showing the negative role of cannabis use in exacerbating mood issues, for those readers who may not be aware of existing scientific support for this. A brief discussio of this would also serve as a counterpoint to the data in this article showing how little the adolescent and youth population understand the risks of cannabis use.  Alternatively, this could have been mentioned briefly in the introduction. This information would have further supported the need for education and prevention efforts aimed at youth using cannabis.

Tables - Table 2 is quite dense; might have benefited from being broken down into two separate tables. 

Author Response

Manuscript ID#: IJERPH-3801705

Corresponding Author: Christopher J. Hammond

Author Responses to Reviewer Comments

Response to Reviewer 1 Comments

1. Summary

Thank you very much for taking the time to review and provide thoughtful feedback on this manuscript. Please find the detailed responses below and the corresponding revisions/corrections highlighted/in track changes in the re-submitted files.

2. Questions for General Evaluation

Reviewer’s Evaluation

Response and Revisions

Does the introduction provide sufficient background and include all relevant references?

Yes

See revisions to introduction on pgs. 2-3.  Also, see Reviewer 1 comment 3.

Are all the cited references relevant to the research?

Yes

Is the research design appropriate?

Yes

See revisions to methods section on pgs. 3-6. Also, see Reviewer 1 comment 5.

Are the methods adequately described?

Yes

See revisions to methods section on pgs. 3-6. Also, see Reviewer 1 comments 4 & 5.

Are the results clearly presented?

Yes

See revisions to results section on pgs. 6-14. See Reviewer 1 comment 6.

Are the conclusions supported by the results?

Yes

See revisions to discussion and conclusion sections on pgs. 14-18. Also, see Reviewer 1 comment 7.

Are all figures and tables clear and well presented

Yes

See revisions to tables. Also see Reviewer 1 Comments 9 and Authors Response to this Comment.

3. Point-by-point response to Comments and Suggestions for Authors

Comments 1: “This well-crafted article addresses an important issue for adolescents and young adults…”

Response 1: Thank you for your positive comment on our manuscript.

Comments 2: Abstract: The abstract is “concise, includes pertinent findings”.

Response 2: Thank you for noting this strength. We agree.

Comments 3: Introduction: The introduction “Describes well what is known about attitudes and expectancies regarding cannabis use and what is known about the role of legalization of either recreational or medical marijuana.” and “Outlines the current aims clearly”.

Response 3: Thank you for noting this strength of the introduction.

Comments 4: Measures: “The authors utilize a survey on marijuana attitudes and beliefs that they have created de-novo. This reviewer would have preferred more information about this survey in the body of the paper, not as a supplement, to include number of items, domain of items, etc.” “Also, there was no attempt to provide even preliminary validation of this instrument (which is commented upon in the Discussion). However, basic psychometric analysis could have been done to provide some sense of its properties and validity.”

Response 4: Thank you for pointing this out. Your comment about providing additional information about the survey instrument and its properties aligns with other reviewer comments (see reviewer 2 comment 6 and reviewer 3 comment 3). Taking your and other reviewer comments in aggregate, we agree with the overarching recommendation to update the manuscript to improve clarity about the survey instrument and its properties and have done so in the revised manuscript. To provide some additional background to reviewers: This is first manuscript we are publishing from the MABS study. Thus, we chose to include a description of the full survey instrument in this manuscript (so that we could cite this manuscript for future MABS study reports), even though we did not statistically analyze and present results from all survey items in current report.  That is why we provided a modest description of the survey in the methods section but included detailed information about all survey items in the supplement.  From your comments, we see how our original description of the survey in methods section provided insufficient detail, especially about the specific items examined in this report. In our revised manuscript we have addressed this, describing the full survey and also making clear which specific survey items were examined in the main text, while retaining the more comprehensive description of the full survey in the supplement.   

In response to your and other reviewer comments about the survey instrument/measure, we have updated/revised the manuscript to provide more information about the survey, its development, item selection, and item sources in the main text.  We agree with your and other reviewer comments that the survey, as a whole instrument is not validated, although several items were taken from other survey instruments. To address this concern we revised the discussion section including this as a study limitation. 

The updated sections of text in the manuscript are presented below with changes depicted in bold:

Updated parts of the Methods Section (Pages 4-5, Lines 186-219):

“2.3.1. Marijuana Attitudes Beliefs and Behaviors Study (MABS) survey instrument  

   The MABS survey was developed for this study from themes elicited from group discussions of child mood disorder experts at NNDC Child and Adolescent Mood Disorder Interest Group meetings focused around better understanding youth, parent, and provider attitudes, beliefs, and behaviors related to cannabis and CBD in the current US healthcare environment and cannabis policy landscape. Survey development was guided by expert discussion around clinically relevant themes for youth mood disorders and informed by a scoping literature review on the topic with items chosen or developed to address existing gaps in the literature. Many survey items were modeled after those used in previous survey studies [28, 31-33] with some taken directly from validated instruments (e.g., risk perception items). The full survey included 129 items querying cannabis and CBD related attitudes and beliefs (5 items), use intentions (3 items), health perception of risk for harm and medical benefit (4-items), perceived accessibility (1 item), perceived use norms (1 item), perceived knowledge about cannabis and CBD use effects (12 items), MC and CBD use behaviors (4 items), expectancies about cannabis, MC, and CBDs’ effects on mood, anxiety, and other psychiatric symptom domains (48-items), along with parent-youth communication and parenting practices (31 items), mental health provider practice behaviors and patient-provider communication about cannabis, MC, and CBD (11 items), demographics (5 items) and clinical characteristics (6 items). The survey used branching logic. After completing demographic and clinical questions, participants receive definitions for MC and CBD and were asked if they had heard of these products.  Participants who had not heard of MC or CBD were instructed to skip related survey items. Parallel youth, parent, and provider versions of the survey were created. See Supplemental methods S1-2 for additional details.

    Items from the MABS survey used as variables in the current report included: demographics (age, sex, parent education, state of residence), clinical/treatment characteristics (lifetime psychiatric symptoms from different domains/categories, current treatment type/setting/modality), cannabis and CBD medical beliefs and attitudes, perceptions about risk for harm and benefits for mental health from regular use of cannabis and CBD, cannabis and CBD use intentions and behaviors.”

Updated parts of the Discussion Section (Page 17, Lines 575-584):

One study limitation is related to measurement validation and content. While we used a formative content validation approach to develop the survey and included items from validated instruments, the full survey has not been psychometrically validated. Notably, some content domains explored in this report related to cannabis outcome perceptions did not have validated instruments at the time of data collection. Development and psychometric validation of instruments that measure use behaviors, attitudes, beliefs, perceptions, and expectancies for different cannabinoid-based products is an important area of research focus for the field that is needed to improve measurement validity for future studies.” 

Comments 5: Statistical Analyses “Discussion is relevant and analytic plan seems robust.”

Response 5: Thank you for noting the strength of our statistical analysis.  

Comments 6: Results: The Results are “Clearly stated and readily understood”.

Response 6: Thank you for noting this strength of the Results.

Comments 7: Discussion: “Overall, the discussion addresses the results directly and is clearly written. Strengths and limitations are addressed.”

Response 7: Thank you for your comment about the discussion section.

Comments 8: Discussion: “Recognizing the word-limit of articles, this [reviewer] still would have liked even a limited mention of the some of the data showing the negative role of cannabis use in exacerbating mood issues, for those readers who may not be aware of existing scientific support for this. A brief discussion of this would also serve as a counterpoint to the data in this article showing how little the adolescent and youth population understand the risks of cannabis use. Alternatively, this could have been mentioned briefly in the introduction. This information would have further supported the need for education and prevention efforts aimed at youth using cannabis.”

Response 8: Thank you for pointing this out.  We agree with this comment.

In response, we have revised both the introduction and discussion sections to include scientific data on the negative effect of cannabis use on mood symptoms in youth for readers who are not aware of the existing scientific support for this. In the revised Discussion section, we have added a paragraph and multiple citations to more clearly frame the study findings showing that adolescents with mood disorders and their parent/caregiver largely believe that cannabis is beneficial for mood problems as being inconsistent with existing scientific data (which shows the opposite). The revised discussion section has also been updated to describe how the study findings point to a need for education and prevention efforts aimed at youth using cannabis. Your suggestion to alternatively consider including this information about cannabis exposure in the introduction section dovetails with feedback from reviewers 2, 3, and 4 each of whom asked for us to provide additional background and references in the introduction. Thus, to be responsive, we have revised both the introduction and discussion sections. 

The updated sections of text in the manuscript are presented below with changes depicted in bold:

Updated Introduction Section (Pages 2-3, Lines 45-116):

    “Rapid changes to United States (US) state-level cannabis laws (CL) over the past 20 years carry unknown implications for youth health. As of November 2024, medical and recreational cannabis are legal in 39 and 24 states and Washington DC, respectively [1]. State-level legislative changes to cannabis’s legal status and federal passage of the 2018 Farm Bill deregulating the production and sale of hemp, a cannabis plant variant containing < 0.3% of delta-9-tetrahydrocannabinol (delta-9-THC) [2], have led to a billion-dollar US cannabinoid market, resulting in increased availability and use of cannabis-derived and hemp-derived cannabinoid products across the US [2]. The main plant-derived cannabinoids from cannabis are delta-9-THC, the main psychoactive constituent responsible for intoxicating effects, and cannabidiol (CBD), a non-psychoactive cannabinoid under investigation for therapeutic properties.  In the US, CBD is available as a prescribed medicine that is approved by the US Food & Drug Administration (FDA) for two rare seizure disorders (Lennox-Gastaut and Dravet Syndrome) in children and tuberous sclerosis in adults; in some cannabis products obtained from medical cannabis (MC) dispensaries and retail outlets in states with CL; and in hemp-derived dietary and health supplements in multiple formulations (e.g., gummies, vapes, tinctures, CBD-infused food, drinks, topicals). Cannabis (used medically or recreationally) and hemp-derived CBD supplements are the most commonly used cannabinoid-based products in the US. There has been growing demand for cannabis and CBD across the US amidst broad claims of purported health benefits that have greatly outpaced the scientific literature substantiating their use as therapeutic agents. Downstream public health effects related to shifting medical cannabis laws (MCL) and recreational cannabis laws (RCL) and the emerging cannabinoid market are unclear and may have both negative and positive outcomes for different subgroups [3, 4]. 

      Youth undergo rapid neurodevelopmental changes making them more prone to experimentation with cannabis and making their brain’s more sensitive to cannabis exposure effects compared to adults [5]. This in combination with the high rates of youth cannabis use [6], makes them a potentially vulnerable subgroup with regard to negative health outcomes related to changing state and federal cannabis policies and expanded societal availability [3, 4, 7-11].  Psychiatric disorders, particularly mood, psychotic, and substance use disorders, can be exacerbated by regular cannabis use, and youth with psychiatric disorders may be more vulnerable to cannabis-related adverse health outcomes, including elevated risk of substance use disorders and poorer course of serious mental illness [3, 7-10, 12].  In the current cannabis policy landscape, youth with mood disorders represent a uniquely vulnerable population that is both more likely to use cannabis and is at elevated risk for experiencing cannabis-related adverse health outcomes. At the same time, rates of CBD use have increased in US youth, and may be higher in youth with mood problems. Given this, research into cannabis and CBD attitudes, medical beliefs, and perceptions and the influence of changing CLs on cannabis and CBD use in youth with mood disorders is warranted [13].

      Cannabis remains the most commonly used federally illicit drug by US adolescents and young adults, with cannabis use disorder being the most common substance use disorder for which US adolescents receive treatment [3].  Over the past two decades, cannabis containing products have become more potent and available, and the perception of risk for harm related cannabis use among US youth in general has decreased, resulting in a more permissive environment and increased access to cannabis products with high potency/concentrations of delta-9-THC [3, 14].  This is concerning given that use of cannabis, particularly high delta-9-THC potency cannabis, is associated with adverse health outcomes, increased risk for psychiatric disorders, academic failure, and higher rates of morbidity and mortality [15, 18, 19].  CBD use has also increased among US youth and adults, driven primarily by increased use of CBD containing dietary/health supplements.  A recent Gallup poll showed that nearly one fifth of US young adults reported using CBD supplements in the past year.  Rates of vaping CBD and using CBD supplements to target health concerns has also increased among US youth.  Many US adults and youth report using CBD supplements for health concerns (often without consulting their doctor) in an attempt to self-medicate mood symptoms, pain, or sleep problems or to replace their conventional medications, despite limited evidence of safety and efficacy.  This is also concerning as a major unanswered question in the field remains whether CBD exposure during critical periods disrupts neurodevelopmental processes in a way similar to delta-9-THC [20]. While generally considered to have a good drug safety profile, CBD products are not risk free.  The use of CBD products may pose risks for side effects, adverse events, and supplement-to-drug interactions with prescribed medications in some patients. Furthermore, several studies have shown that commonly available CBD supplements are often mislabeled, contaminated, or adulterated, and may contain trace amounts of THC in sufficient doses to produce psychoactive effects or result in positive urine drug test results for cannabis.  Thus, the risks related to CBD use may be amplified and the potential benefits may be more difficult to measure for patients who use CBD supplements compared to pharmaceutical CBD medications. In light of this, studies investigating real-world use of different cannabis chemotypes and CBD products in youth are needed.” 

Updated Discussion Section (Page 15-16, Lines 438-492):

     “Results from this study show that US youth receiving mood disorder treatment and their parents perceive cannabis and CBD products to be safe and effective treatments for multiple pediatric mental health conditions. Most youth and parents in the sample reported believing that regular use of cannabis and CBD products reduce depression and anxiety, with nearly half of youth and one-third of parent respondents believing cannabis use to be beneficial for suicidal thoughts and behaviors, in direct contrast to the current evidence [9, 38].  Additionally, one-in-five youth respondents reported believing that cannabis is beneficial in treating psychosis, which is concerning in light of research showing links between cannabis use and psychosis [18]. Our results are consistent with previous studies showing trends towards more positive cannabis related attitudes and medical beliefs, elevated perception of health benefits, reduced perception of risk for harm, and increased use rates across the entire US adult population and subgroups of US adolescents and young adults [20, 22-24, 26].  Additionally, our youth findings are consistent with prior studies on CBD attitudes in young adults and individuals with substance use disorders.  Our parent findings align with a recent study by Schwaller and colleagues that found positive safety perceptions about unlicensed CBD products and high administration rates of these products to children with ADHD, autism spectrum disorder, and anxiety disorder by US parents who had a 7-18 year-old child diagnosed with those conditions [54].  Our attitudinal results also parallel findings from international studies which have shown shifting societal attitudes, policies, and use patterns related to both cannabis and CBD in many countries around the world. Our study findings build off of national survey studies showing that trends towards more favorable attitudes and perceptions about cannabinoids are also occurring in vulnerable subgroups, in this case, youth with mood disorders.  These findings are of particular importance, given the current mental health crisis in US youth and the US mental healthcare system’s struggle to respond to this crisis, made more complicated by the rapid growth of multibillion-dollar cannabis and CBD product markets which have promoted broad claims of health benefits unsupported by the scientific literature [3, 39].  Recent meta-analyses and longitudinal studies show that cannabis use in adolescents with and without mood disorders is associated with poorer mood outcomes, reduced treatment response in those receiving mental health care and increased risk for depression, suicidal ideations, and suicide attempts by young adulthood [9, 38]. Concerningly, regular cannabis use and cannabis use disorders may increase mortality in young people with mood disorders.  A recent study by Fontanella et al. (2021) that examined Ohio Medicaid Claims data from 204,780 US youth (ages 10-24) found that, among youth with mood disorders, having a cannabis use disorder diagnosis was associated with increased non-fatal self-harm and all-cause mortality.  In contrast to the evidence for harm from cannabis use, the evidence for benefit from cannabinoid-based therapeutics for psychiatric conditions remains in its infancy. Safety and efficacy data on CBD (and other cannabinoids) as therapeutic agents for mood disorders remains limited, especially in youth samples; although early studies have shown promise for select conditions (e.g., anxiety), and multiple clinical trials are underway. A recent systematic review by Hasbi et al. (2023) examined the therapeutic role of cannabinoids including THC and CBD for psychiatric conditions and concluded that the evidence that cannabinoids improve depression or anxiety disorders is currently weak and of low-quality [40].  In sum, there is currently moderate to strong scientific evidence showing elevated risk for negative health outcomes from cannabis exposure in youth with and without mood disorders and insufficient evidence for safety and efficacy of cannabis or CBD products for pediatric neuropsychiatric conditions beyond rare treatment resistant seizure disorders.  Taken within this context, our attitudinal findings point to low cannabinoid health literacy and significant discrepancies between what youth with mood disorders and their parents believe about the risks and benefits of cannabis and CBD use and the current evidence related to safety and efficacy of these products, particularly in relation to cannabis’s effects on mental health and the therapeutic use of CBD and other cannabinoids for the treatment of mood disorders, psychotic disorders, and suicidal behaviors.”

Comments 9: Tables and Figures: “Table 2 is quite dense; might have benefited from being broken down into two separate tables.”

Response 9: We agree. In response to this comment, we broke down Table 2 into two separate tables one focused on cannabis related attitudes and perceptions (new Table 2) and the other focused on CBD related attitudes and perceptions (new Table 3). The updated tables are shown on pgs. 10-12 of the revised manuscript.    

4. Response to Comments on the Quality of English Language

No comments made on the quality of English language

Response: N/A

5. Additional clarifications

N/A

Reviewer 2 Report

Comments and Suggestions for Authors

This paper examines variations in attitudes towards the use of cannabis and cannabidiol (CBD) in youth with mood disorders and their parents. The stated aim of this research is to examine variations according to the legality of these substance in each state of the United States.

In view of the recent legalization of these substances in several states, and their complex effects on the course and outcome of mood disorders, there is a clear rationale for the current study. The authors have used anonymous surveys to reduce the risk of social desirability bias, and have interviewed both youth and their parents, thereby assessing for intergenerational and patient-caregiver variations. Multivariate and sensitivity analyses have been conducted to examine the strength, consistency, and robustness of reported associations. Overall, this paper is methodologically sound and addresses a topic of clinical and public health relevance, and I commend the authors for undertaking this work.

There are certain aspects of the manuscript which would benefit from correction or clarification, as appropriate. These are enumerated below:

  1. Abstract: It would be helpful to mention, even if only briefly, the rationale for selecting youth with mood disorder. CBD is used to medicate (or self-medicate) several conditions, including mood, anxiety and chronic pain disorders, and cannabis may trigger episodes or worsen the course of some mood disorders (most specifically bipolar disorders). While these are good reasons for studying attitudes towards cannabis and CBD in this patient group, this should be stated clearly.
  2. Introduction: Paragraph 2 (lines 59-68) should be expanded slightly, to address two concerns. The first is the same one flagged in the Abstract - why should we be concerned about mood disorders in particular? The second, which is linked to the first, is that more recent and systematic information on the links between cannabis, CBD and mood disorders should be presented here, at least in summary format.
  3. Methodology - General: As this is an observational study, please mention adherence to the STROBE guidelines, and - if possible - submit a completed STROBE checklist as part of the supplementary material.
  4. Participants and procedures: The inclusion and exclusion criteria should be reported in more detail. Were youth with comorbidities that could influence their attitudes to cannabis / CBD (e.g., ADHD, conduct disorder) excluded or included? Were youth with a current or past history of cannabis (or other drug) misuse included? Did any of the youth have medical comorbidities (e.g., chronic pain, epilepsy) for which cannabis / CBD are sometimes advocated as treatments? Likewise, were the parents screened for mental illness or substance use disorder, and did this influence their inclusion in the study?
  5. Study measures: The authors have stated that the MABS survey was developed specifically for this study. Was it tested in a pilot population prior to its use in this study - or is the current study intended to be a pilot? Were any other methods (e.g., content validation by subject experts / other stakeholders) used to improve the validity of the survey? What are its psychometric properties? (If this data is not available, this could be acknowledged as a limitation of the current research, or an avenue for future work in this field.)
    Was any tool used to assess patients' current levels of mood and anxiety symptoms? This could significantly influence their attitudes towards the use of cannabis and CBD (e.g., an adolescent with hypomanic symptoms might be more willing to experiment with the former; one with anxiety might attempt to self-medicate with the latter).
  6. Statistical analysis: Were other covariates such as age of onset, duration of mood disorder, presence of psychiatric or medical comorbidities, or substance use history considered for inclusion in the multivariate analyses? If not, why? (Some of these appear to be mentioned in Section 3.5 of the results, but they are not mentioned in the Methods or Statistical Analysis sections.)
  7. Results: Was there any correlation between youth and parental attitudes towards cannabis and CBD in those families where both youth and parents participated? If so, what could this imply?
  8. Discussion: This section needs to be expanded slightly in two domains:
    a) A more in-depth appraisal of the study results. The Results section presents several variables that could be linked to attitudes towards cannabis or CBD, and / or readiness to use these substances. What are the clinical or public health implications of these findings (e.g., the ones presented in Table 2 and Table 3)?
    b) Comparisons with similar studies in other relevant populations, or with general population studies of youth. Even if this is the first study to  specifically examine mood disorders, it would be interesting to contrast the current findings with those seen in young adults in general (e.g., Wheeler et al., 2020; Nguyen et al., 2023) or in people with substance use disorders (e.g., Kudrich et al., 2024). 
  9. Conclusions: Were any participants aware of the potential risks / harms associated with cannabis (not CBD) use in mood disorders? This could be highlighted after lines 122-123.

Author Response

Manuscript ID#: IJERPH-3801705

Corresponding Author: Christopher J. Hammond

Author Responses to Reviewer Comments

Response to Reviewer 2 Comments

1. Summary

Thank you very much for taking the time to review and provide feedback on our manuscript. Please find the detailed responses below and the corresponding revisions/corrections highlighted/in track changes in the re-submitted files.

2. Questions for General Evaluation

Reviewer’s Evaluation

Response and Revisions

Does the introduction provide sufficient background and include all relevant references?

Can be improved

See revisions to introduction section on pgs. 2-3.  

Are all the cited references relevant to the research?

Yes

Is the research design appropriate?

Yes

Are the methods adequately described?

Can be improved

See revisions to methods section on pgs. 3-6.

Are the results clearly presented?

Yes

Are the conclusions supported by the results?

Can be improved

See revisions to discussion and conclusions sections on pgs. 14-18.

Are all figures and tables clear and well presented

Yes

3. Point-by-point response to Comments and Suggestions for Authors

Comments 1:  “In view of the recent legalization of these substances in several states, and their complex effects on the course and outcome of mood disorders, there is a clear rationale for the current study.”… “The authors have used anonymous surveys to reduce the risk of social desirability bias, and have interviewed both youth and their parents, thereby assessing for intergenerational and patient-caregiver variations. Multivariate and sensitivity analyses have been conducted to examine the strength, consistency, and robustness of reported associations.”…”Overall, this paper is methodologically sound and addresses a topic of clinical and public health relevance, and I commend the authors for undertaking this work.”

Response 1: Thank you for your thoughtful feedback and detailed comments on our manuscript including the comments noted above that reference the clear study rationale, methodological rigor, and significance of the topic of the manuscript.  

Comments 2: Abstract: “It would be helpful to mention, even if only briefly, the rationale for selecting youth with mood disorder. CBD is used to medicate (or self-medicate) several conditions, including mood, anxiety and chronic pain disorders, and cannabis may trigger episodes or worsen the course of some mood disorders (most specifically bipolar disorders). While these are good reasons for studying attitudes towards cannabis and CBD in this patient group, this should be stated clearly.”

Response 2: We agree. In response, we have modified the abstract to provide additional background on why cannabis and CBD related attitudes, medical beliefs, and health benefit and risk perceptions are relevant in youth with mood disorders.

The updated section of the abstract is presented below with changes depicted in bold:

Updated Abstract (Page 1, Lines 25-27):

“Dramatic shifts in state-level cannabis laws (CL) and federal hemp regulations have resulted in increased availability and use of cannabis and cannabidiol (CBD) products throughout the US with unknown implications for youth. Youth with mood disorders represent a vulnerable population that is more likely to use cannabis and CBD and is at elevated risk for experiencing cannabis-related adverse health outcomes. This multisite study characterized attitudes, health perceptions, and behaviors related to cannabis and CBD use among US youth receiving mood disorder treatment and their parents and assessed whether attitudinal differences varied as a function of respondent group and state-level CL status. ….”

Author comment related to abstract word count: We wanted to point out that the original version of the abstract (200 words) met the 200 word limit for IJERPH abstracts based on the author guidelines.  With the additional sentence added in response to reviewer 1 comment 2, the revised abstract is now at 228 words. We were not sure how important it was to keep the abstract at/below the 200 word limit and can edit it further based on the editor and reviewer’s feedback if required.

Comments 3: Introduction: “Paragraph 2 (lines 59-68) should be expanded slightly, to address two concerns. The first is the same one flagged in the Abstract - why should we be concerned about mood disorders in particular? The second, which is linked to the first, is that more recent and systematic information on the links between cannabis, CBD and mood disorders should be presented here, at least in summary format.”

Response 3: We agree. In response, we have modified paragraph 2 of the introduction to provide additional background on why cannabis and CBD related attitudes, medical beliefs, and health benefit and risk perceptions are relevant in youth with mood disorders and additional background summarizing the current scientific literature on the links between cannabis, CBD, and mood disorders.  As part of this revision, we have also added new citations to the manuscript. 

The updated introduction section is presented below with changes depicted in bold:

Updated parts of the Introduction Section (Pages 2-3, Lines 46-116):

    “Rapid changes to United States (US) state-level cannabis laws (CL) over the past 20 years carry unknown implications for youth health. As of November 2024, medical and recreational cannabis are legal in 39 and 24 states and Washington DC, respectively [1]. State-level legislative changes to cannabis’s legal status and federal passage of the 2018 Farm Bill deregulating the production and sale of hemp, a cannabis plant variant containing < 0.3% of delta-9-tetrahydrocannabinol (delta-9-THC) [2], have led to a billion-dollar US cannabinoid market, resulting in increased availability and use of cannabis-derived and hemp-derived cannabinoid products across the US [2]. The main plant-derived cannabinoids from cannabis are delta-9-THC, the main psychoactive constituent responsible for intoxicating effects, and cannabidiol (CBD), a non-psychoactive cannabinoid under investigation for therapeutic properties.  In the US, CBD is available as a prescribed medicine that is approved by the US Food & Drug Administration (FDA) for two rare seizure disorders (Lennox-Gastaut and Dravet Syndrome) in children and tuberous sclerosis in adults; in some cannabis products obtained from medical cannabis (MC) dispensaries and retail outlets in states with CL; and in hemp-derived dietary and health supplements in multiple formulations (e.g., gummies, vapes, tinctures, CBD-infused food, drinks, topicals). Cannabis (used medically or recreationally) and hemp-derived CBD supplements are the most commonly used cannabinoid-based products in the US. There has been growing demand for cannabis and CBD across the US amidst broad claims of purported health benefits that have greatly outpaced the scientific literature substantiating their use as therapeutic agents. Downstream public health effects related to shifting medical cannabis laws (MCL) and recreational cannabis laws (RCL) and the emerging cannabinoid market are unclear and may have both negative and positive outcomes for different subgroups [3, 4]. 

      Youth undergo rapid neurodevelopmental changes making them more prone to experimentation with cannabis and making their brain’s more sensitive to cannabis exposure effects compared to adults [5]. This in combination with the high rates of youth cannabis use [6], makes them a potentially vulnerable subgroup with regard to negative health outcomes related to changing state and federal cannabis policies and expanded societal availability [3, 4, 7-11].  Psychiatric disorders, particularly mood, psychotic, and substance use disorders, can be exacerbated by regular cannabis use, and youth with psychiatric disorders may be more vulnerable to cannabis-related adverse health outcomes, including elevated risk of substance use disorders and poorer course of serious mental illness [3, 7-10, 12].  In the current cannabis policy landscape, youth with mood disorders represent a uniquely vulnerable population that is both more likely to use cannabis and is at elevated risk for experiencing cannabis-related adverse health outcomes. At the same time, rates of CBD use have increased in US youth, and may be higher in youth with mood problems. Given this, research into cannabis and CBD attitudes, medical beliefs, and perceptions and the influence of changing CLs on cannabis and CBD use in youth with mood disorders is warranted [13].

      Cannabis remains the most commonly used federally illicit drug by US adolescents and young adults, with cannabis use disorder being the most common substance use disorder for which US adolescents receive treatment [3].  Over the past two decades, cannabis containing products have become more potent and available, and the perception of risk for harm related cannabis use among US youth in general has decreased, resulting in a more permissive environment and increased access to cannabis products with high potency/concentrations of delta-9-THC [3, 14].  This is concerning given that use of cannabis, particularly high delta-9-THC potency cannabis, is associated with adverse health outcomes, increased risk for psychiatric disorders, academic failure, and higher rates of morbidity and mortality [15, 18, 19].  CBD use has also increased among US youth and adults, driven primarily by increased use of CBD containing dietary/health supplements.  A recent Gallup poll showed that nearly one fifth of US young adults reported using CBD supplements in the past year.  Rates of vaping CBD and using CBD supplements to target health concerns has also increased among US youth.  Many US adults and youth report using CBD supplements for health concerns (often without consulting their doctor) in an attempt to self-medicate mood symptoms, pain, or sleep problems or to replace their conventional medications, despite limited evidence of safety and efficacy.  This is also concerning as a major unanswered question in the field remains whether CBD exposure during critical periods disrupts neurodevelopmental processes in a way similar to delta-9-THC [20]. While generally considered to have a good drug safety profile, CBD products are not risk free.  The use of CBD products may pose risks for side effects, adverse events, and supplement-to-drug interactions with prescribed medications in some patients. Furthermore, several studies have shown that commonly available CBD supplements are often mislabeled, contaminated, or adulterated, and may contain trace amounts of THC in sufficient doses to produce psychoactive effects or result in positive urine drug test results for cannabis.  Thus, the risks related to CBD use may be amplified and the potential benefits may be more difficult to measure for patients who use CBD supplements compared to pharmaceutical CBD medications. In light of this, studies investigating real-world use of different cannabis chemotypes and CBD products in youth are needed.” 

Comments 4: Methods: “As this is an observational study, please mention adherence to the STROBE guidelines, and - if possible - submit a completed STROBE checklist as part of the supplementary material.”

Response 4: Thank you for pointing this out.

We have updated the manuscript to describe how we adherence to STROBE guidelines for observational studies; adding a sentence about following STROBE reporting guidelines and including the STROBE guidelines citation. In addition, we have added a completed STROBE checklist to the supplementary materials (eTable S10 on pages 43-45 in the supplemental materials).   

The updated part of the Methods section is presented below with changes depicted in bold:

Updated Methods Section (Page 4, Lines 149-150):

Strengthening the Reporting of Observational Studies in Epidemiology (STROBE) reporting guideline for cross-sectional studies was used [42].”

Comments 5: Methods: Participants and Procedures: “The inclusion and exclusion criteria should be reported in more detail. Were youth with comorbidities that could influence their attitudes to cannabis / CBD (e.g., ADHD, conduct disorder) excluded or included? Were youth with a current or past history of cannabis (or other drug) misuse included? Did any of the youth have medical comorbidities (e.g., chronic pain, epilepsy) for which cannabis / CBD are sometimes advocated as treatments? Likewise, were the parents screened for mental illness or substance use disorder, and did this influence their inclusion in the study?”

Response 5: We agree and have revised the Participants and Procedures section to provide additional details about inclusion/exclusion criteria for the study. To provide clarification about the sampling strategy and inclusion/exclusion criteria: We made a design choice to recruit a sample that was highly generalizable to youth presenting for mood disorder treatment.  To do this we used a broad sampling strategy with highly inclusive inclusion criteria and few exclusion criteria.  As noted in the revised description of the participants on pg. 4, lines 167-70, “Broad inclusion criteria and few exclusion criteria were used to optimize generalizability by allowing participation by a diverse patient population representative of real-world patients from youth mood clinic settings.”  Thus, youth receiving mood disorder treatment at recruitment sites were not excluded if they had psychiatric comorbidities, substance use, or medical comorbidities. As part of the study, we did collect detailed information from youth and parents about the youth’s lifetime mental health symptoms/disorders from different domains/categories as well as mental health treatment types/modalities/settings. However, your question about “Did any of the youth have medical comorbidities (e.g., chronic pain, epilepsy) for which cannabis / CBD are sometimes advocated as treatments?”  highlights a study limitation, as we did not collect information about medical comorbidities as part of the study. In response, we have revised the discussion section of the manuscript to highlight limitations related to our sampling strategy and the absence of data on medical comorbidities, particularly medical comorbidities from conditions that are sometimes treated with cannabinoid-based medicines or CBD. 

The updated sections of text in the manuscript are presented below with changes depicted in bold:

Updated parts of the Methods Section (Page 4, Lines 153-170):

“2.2 Participants and Procedures

2.2.1. Participants

 Adolescents (ages 12-17) and young adults (ages 18-25) receiving treatment at youth mood disorder clinics in six states and parents/caregivers (henceforth referred to as parents) of youth receiving mood disorder treatment were invited to complete an anonymous survey about cannabis and CBD related attitudes and behaviors.

2.2.2. Study Eligibility

For youth, inclusion criteria included ages 12-25 years and currently receiving mental health treatment at one of the six participating youth mood disorder clinics. For parents, inclusion criteria included being a primary caregiver of an adolescent who was currently receiving mental health treatment at the clinics. Exclusion criteria for both respondent groups included lack of English language fluency and cognitive deficits or acute severe mental illness that would interfere with their ability to safely participate in study procedures, understand item content, and complete the survey. Parent and youth participants were not limited to parent-child dyads. Broad inclusion criteria and few exclusion criteria were used to optimize generalizability by allowing participation by a diverse patient population representative of real-world patients from youth mood clinic settings.”

Updated parts of the Discussion Section (limitations subsection) (Pages 17-18, Lines 584-591 and Lines 600-605):

Regarding missing content, we did not collect detailed information on patterns of non-medical/recreational cannabis use, limiting our ability to examine associations between cannabis use and attitudinal factors, although we did collect data on youth alcohol/drug use problems and controlled for this in sensitivity analyses. We also did not collect information about medical comorbidities in youth, and, thus, were unable to control for potential confounding effects from comorbid medical conditions (e.g., chronic pain, cancer, and seizure disorders) that MC and CBD are sometimes used to treat.” …   “It is important to note that the sample is a convenience sample. All youth respondents were receiving mental health treatment in child mood clinics linked to US academic medical centers. Cannabis and CBD related attitudes, beliefs, perceptions, and behaviors from this population may not generalize to non-treatment engaged youth with mood disorders or those receiving treatment in different settings/contexts.”

Comments 6: Methods: Study Measures: “The authors have stated that the MABS survey was developed specifically for this study. Was it tested in a pilot population prior to its use in this study - or is the current study intended to be a pilot? Were any other methods (e.g., content validation by subject experts / other stakeholders) used to improve the validity of the survey? What are its psychometric properties? (If this data is not available, this could be acknowledged as a limitation of the current research, or an avenue for future work in this field.)”

Response 6: Thank you for your comments and questions about the MABS survey instrument. To clarify: We did go through a content validation process with subject experts and child mental health providers (psychiatrists and psychologists) who were a part of the NNDC Child and Adolescent Mood Disorder Interest Group (CAMDIG) to improve its content validity. We did not formally pilot test the survey, although we did administer it to 3-4 undergraduate research assistants and got their feedback on it prior to finalizing the measure its use in the study.  We did not assess the survey’s psychometric properties, although several survey items were from other published surveys validated in youth samples.  Your comment parallels reviewer 1 comment 4 and reviewer 3 comment 3. Please see our responses 1.4 and 3.3 for additional details.   

In response to your comment and other reviewer comments, we have revised the methods section to provide additional information about the survey, its development, item selection, and item sources in the main text. Based on your suggestion, we have also modified the discussion section to acknowledge the lack of psychometric validation of the full survey instrument as a limitation for the study. 

The updated sections of text in the manuscript are presented below with changes depicted in bold:

Updated Methods Section (Pages 4-5, Lines 186-215):

“2.3.1. Marijuana Attitudes Beliefs and Behaviors Study (MABS) survey instrument  

   The MABS survey was developed for this study from themes elicited from group discussions of child mood disorder experts at NNDC Child and Adolescent Mood Disorder Interest Group meetings focused around better understanding youth, parent, and provider attitudes, beliefs, and behaviors related to cannabis and CBD in the current US healthcare environment and cannabis policy landscape. Survey development was guided by expert discussion around clinically relevant themes for youth mood disorders and informed by a scoping literature review on the topic with items chosen or developed to address existing gaps in the literature. Many survey items were modeled after those used in previous survey studies [28, 31-33] with some taken directly from validated instruments (e.g., risk perception items). The full survey included 129 items querying cannabis and CBD related attitudes and beliefs (5 items), use intentions (3 items), health perception of risk for harm and medical benefit (4-items), perceived accessibility (1 item), perceived use norms (1 item), perceived knowledge about cannabis and CBD use effects (12 items), MC and CBD use behaviors (4 items), expectancies about cannabis, MC, and CBDs’ effects on mood, anxiety, and other psychiatric symptom domains (48-items), along with parent-youth communication and parenting practices (31 items), mental health provider practice behaviors and patient-provider communication about cannabis, MC, and CBD (11 items), demographics (5 items) and clinical characteristics (6 items). The survey used branching logic. After completing demographic and clinical questions, participants receive definitions for MC and CBD and were asked if they had heard of these products.  Participants who had not heard of MC or CBD were instructed to skip related survey items. Parallel youth, parent, and provider versions of the survey were created. See Supplemental methods S1-2 for additional details.

    Items from the MABS survey used as variables in the current report included: demographics (age, sex, parent education, state of residence), clinical/treatment characteristics (lifetime psychiatric symptoms from different domains/categories, current treatment type/setting/modality), cannabis and CBD medical beliefs and attitudes, perceptions about risk for harm and benefits for mental health from regular use of cannabis and CBD, cannabis and CBD use intentions and behaviors.”

Updated Discussion Section (Page 17, Lines 575-584):

One study limitation is related to measurement validation and content. While we used a formative content validation approach to develop the survey and included items from validated instruments, the full survey has not been psychometrically validated. Notably, some content domains explored in this report related to cannabis outcome perceptions did not have validated instruments at the time of data collection. Development and psychometric validation of instruments that measure use behaviors, attitudes, beliefs, perceptions, and expectancies for different cannabinoid-based products is an important area of research focus for the field that is needed to improve measurement validity for future studies.”

Comments 7: Methods: Study Measures: “Was any tool used to assess patients' current levels of mood and anxiety symptoms? This could significantly influence their attitudes towards the use of cannabis and CBD (e.g., an adolescent with hypomanic symptoms might be more willing to experiment with the former; one with anxiety might attempt to self-medicate with the latter).”

Response 7: Thank you for your question about our study measures and comment about influence of mood symptoms on attitudes. We’ve responded to them point-by-point below.

Comment 7.1: “Was any tool used to assess patients' current levels of mood and anxiety symptoms?”:

Response 7.1: Yes, we used brief validated instruments to assess each youth’s current level of depression and anxiety symptoms. Specifically, all youth respondents completed the Patient Health Questionnaire-2-item version Depression scale (PHQ-2) and the Generalized Anxiety Disorder-2-item Anxiety scale (GAD-2) as part of the study.  See Methods Sections 2.3.3 and 2.3.4 (pg. 5, lines 222-236) for additional details about these measures. Current levels of depression and anxiety symptoms (mean PHQ-2 and GAD-2 scores) for youth respondents are presented in Table 1.

Comment 7.2: “This could significantly influence their attitudes towards the use of cannabis and CBD (e.g., an adolescent with hypomanic symptoms might be more willing to experiment with the former; one with anxiety might attempt to self-medicate with the latter).”

Response 7.2: We agree. As part of the study we conducted two sets of analyses investigating the influence of current depressive and anxiety symptoms on cannabis and CBD attitudes and use intentions.  First, as part of our sensitivity analyses we sought to test the influence of current anxiety and depression on our main analysis (examining group differences in cannabis and CBD attitudes).  From Methods Section, 2.6 Sensitivity Analyses (pg. 6, lines 262-276): “Third, we reran [our main] analyses in youth with current depression and anxiety disorders (i.e., PHQ-2 or GAD-2 >3) to determine impact of current depression or anxiety on previously established associations.” Second, one of our main study objectives (pg.3, lines 139-141) was “to identify whether individual-level factors (e.g., demographic and clinical characteristics, personal attitude/expectancies), and state-level CL status were associated with youth intention to use cannabis and CBD”.  To do this “we ran ordinal regression models for each factor, covarying age, sex, parent education, and site. Then, we used multivariable ordinal regression analyses to characterize predictors of use intentions, incorporating each factor that significantly correlated with cannabis and CBD use intentions to determine whether those factors were associated with intentions to use after controlling for other relevant variables.” (from pg. 6, lines 257-261).  As part of those analyses we examined PHQ-2 and GAD-2 scores as factors of interest.   

Comments 8: Methods: Statistical Analysis: “Were other covariates such as age of onset, duration of mood disorder, presence of psychiatric or medical comorbidities, or substance use history considered for inclusion in the multivariate analyses? If not, why? (Some of these appear to be mentioned in Section 3.5 of the results, but they are not mentioned in the Methods or Statistical Analysis sections.).”

Response 8: Thank you for your questions about covariates and analysis.

To clarify covariate selection: age, sex, parent education, and site were included as covariates of no interest in all of our statistical models. These variables were selected as covariates to ensure that we controlled for variation in results based on age, sex, socioeconomic status (using parent education as a proxy) and site. They are standard covariates commonly used in this type of analysis to control for confounding related to sociodemographic characteristics, and were deemed to be sufficient for our main analyses by the statistical experts. We also performed several sensitivity analyses to ensure robustness of our main results and to test whether differences in cannabis and CBD-related medical beliefs, attitudes, perceptions, and expectancies related to respondent group and state ML status varied as a function of other measured and unmeasured factors. These sensitivity analyses were conducted specifically to determine whether our main results varied as a function of site, parent-youth dyad, current depression and anxiety disorders, and lifetime history of alcohol or drug use problems. In addition, we also reran our respondent group analyses covarying for total psychiatric problems as youth and parent respondents differed on this measure.

In response to your question ”Were other covariates such as age of onset, duration of mood disorder, presence of psychiatric or medical comorbidities, or substance use history considered for inclusion in the multivariate analyses?”  

Regarding age of onset, duration, presence of medical comorbidities: Our survey did not collect information on age of onset of mood disorder, duration of mood disorder, or medical comorbidities, and thus these factors could not be examined as covariates. As described above in our response to comment 5, we have added text to the discussion section (limitations subsection) noting this missing survey content as a limitation of the study.

Regarding psychiatric comorbidities: Yes, we considered psychiatric comorbidities as a covariate.  As noted in our response to comment 5 above, as part of the study we collected information on psychiatric comorbidities and reported on them as part of our descriptive and comparative analyses. The proportion of youth from the sample reporting lifetime psychiatric symptoms/problems from each symptom category is presented in Table 1. We also created a total psychiatric problem score to index total psychiatric problems for each participant (calculated by coding each symptom category as 1 = present and 0 = absent and summing across categories). This score is presented in Table 1.  Psychiatric symptom/problems from each category and the total psychiatric problem score were included as factors of interest in our predictors of intentions to use cannabis analyses.

After finding differences between youth and parents in total psychiatric problem scores, we conducted additional post-hoc sensitivity analyses re-running our main respondent group analyses covarying for total psychiatric problems.  This is presented in the Results section (pg. 9, lines 343-344) where we note “Results were unchanged when analyses were rerun covarying for total psychiatric problems.”

Regarding substance use history: Yes, we considered substance use history as a covariate, examining the influence of substance use history on our results. This was conducted as part of our sensitivity analyses. Specifically, we reran our main analysis in a subgroup of youth with no lifetime history of alcohol or drug problems.

Based on your comment here, we have made minor text edits to the Methods section to improve clarity. These text edits are focused on ensuring that the article more clearly describes the analyses, covariates used, and sensitivity analyses and post-hoc analyses conducted as part of this study.

The updated sections of text in the manuscript are presented below with changes depicted in bold:

Updated Methods Section (Page 6, Lines 243-276):

2.5. Statistical analysis 

We used IBM SPSS Statistical Analysis Software v25.0 (IBM, Armonk, NY); significance was set at two-sided p<0.05. Discrete missing values were minimal (0% to 3.0%) and imputed using mean substitution methods. Results were summarized using descriptive statistics to characterize beliefs, attitudes, perceptions, and expectancies, for cannabis and CBD in the total sample and across respondent groups. Multinomial logistic and linear regression analyses with state-level CL status (RCL vs. MCL vs. no-CL) and respondent group (youth vs. parent) as between-group factors tested for state-level CL and intergenerational differences. We ran separate post-hoc covariate adjusted models for group and CL status effects in the total sample, then examined CL status effects separately for youth and parents to parse group-by-CL-status interactions. Age, sex, parent education, and site were included as covariates of no interest in the models. We used two-step analysis to investigate demographic, clinical, attitudinal, and state CL factors that were associated with intentions to use cannabis and CBD in the youth respondent group.  As a first step, to reduce data, we ran ordinal regression models for each factor, covarying age, sex, parent education, and site. Then, we used multivariable ordinal regression analyses to characterize predictors of use intentions, incorporating each factor that significantly correlated with cannabis and CBD use intentions to determine whether those factors were associated with intentions to use after controlling for other relevant variables.

2.6. Sensitivity analyses

Four sensitivity analyses were performed to ensure robustness of main results and to test whether differences related to group and CL status varied as a function of other measured and unmeasured factors. First, we applied leave-one-out validation (LOOV), holding out/censoring respondents from each site and running main analytic models in the n - 1 sample to ensure results were not being driven by a single site. Second, we reran analyses in a subgroup of matched youth-parent dyads (30 respondents/group). This test enabled us to examine differences in beliefs, attitudes, perceptions, and expectancies in relation to respondent group and CL status in a sample with limited between-group variation from measured and unmeasured environmental factors.  Third, we reran analyses in youth with current depression and anxiety disorders (i.e., PHQ-2 or GAD-2 >3) to determine impact of current depression or anxiety meeting clinical/diagnostic thresholds influenced the associations identified in our main analyses. Finally, we reran analyses for youth with no lifetime history of alcohol or drug use problems (75 youth, 57 parent report). 

Comments 9: Results: “Was there any correlation between youth and parental attitudes towards cannabis and CBD in those families where both youth and parents participated? If so, what could this imply?”

Response 9: Thank you for this thoughtful question.  We did not conduct analyses examining correlations between youth and parent attitudes towards cannabis and CBD in youth-parent dyads.  Similar to reviewer 2, our team is very interested in this question. However, we also feel strongly that there is insufficient space in the current manuscript to appropriately address the question about concordance with the focus and nuance that it deserves and that the question is beyond the scope of the current report. We believe that this topic warrants a separate manuscript as it represents a separate but related knowledge gap in this field that carries unique clinical relevance for parent-youth dyadic prevention efforts.  As such, we plan to investigate this question as part of a follow-up article focused on investigating the degree of concordance vs discordance in attitudes, beliefs, and behaviors between matched youth and parent respondents (i.e., youth-parent dyads) from the sample.

We do want to highlight for the reviewer, that while we did not investigate correlations between matched youth and parents as part of this manuscript, we did examine matched youth-parent dyads as part of our sensitivity analyses. Specifically, we reran our main analyses in a subgroup of matched youth-parent dyads (30 respondents/group).  This sensitivity test was performed with the goal of enabling us to examine differences in beliefs, attitudes, perceptions, and expectancies in relation to respondent group and CL status in a sample that may have lower variance in measured and unmeasured environmental (confounding) factors on account of them being matched youth and parent respondents, many of whom are living in the same household.  Those sensitivity analyses are described on pg. 6, lines 267-271 and pg. 14, lines 427-430.

Comments 10: Discussion: “This section needs to be expanded slightly in two domains:

a) A more in-depth appraisal of the study results. The Results section presents several variables that could be linked to attitudes towards cannabis or CBD, and / or readiness to use these substances. What are the clinical or public health implications of these findings (e.g., the ones presented in Table 2 and Table 3)?”

Response 10: We agree. In response, we have modified the discussion to provide a more in-depth appraisal of the study findings within the context of the larger literature and to further explore the clinical relevance and public health implications of the study findings. We have also added new citations to this section to further strengthen it.  

The updated section of the Discussion is presented below with changes depicted in bold:

Updated Discussion (Page 17, Lines 545-572):

Based on our study findings, goals of next-stage multilevel prevention efforts should focus on cannabinoid health literacy, targeting youth and parent misperceptions, providing education about mental health risks related to cannabis use; while concurrently promoting aggressive treatment of youth mood disorders to reduce the likelihood these youth will initiate cannabis or progress to regular use. Attitudinal and mental health factors represent important and malleable intervention targets and can inform public health cannabis messaging efforts [59, 75], especially as youth living in RCL states have more favorable cannabis attitudes associated with increased intention to use.  Cannabis prevention efforts for this population should largely be driven by pediatric medical and mental health clinicians; public health campaigns should focus on cannabis health literacy by providing targeted, evidence-based education to youth and parents and encouraging fact-driven discussions between parents, youth, and providers about cannabinoids and mental health.  Standardized medical/healthcare education and workforce training for pediatric mental and medical healthcare providers about cannabinoid risks and benefits and how to screen for cannabinoid product use and advise youth/families are needed. In order to be effective, this medical/healthcare education on cannabis and cannabinoids should be included as required training content for residency/fellowship programs and continued medical education content for board certification and maintenance for psychiatrists, psychologists, pediatricians, primary care providers, and other licensed mental health clinicians.  Importantly, taking lessons from the history of alcohol and tobacco industry’s influence on medical/healthcare practice [77, 78], medical education content on cannabis and cannabinoids should be unbiased and evidence-informed, and consider and weight the risks and potential benefits from different cannabinoid products.  This educational content should be prepared and disseminated by experts in cannabis prevention and early intervention research without cannabis industry influence. Furthermore, future cannabis regulatory and policy reforms should incorporate strong evidence-informed risk mitigating provisions and resources that seek to reduce cannabis use in youth and protect vulnerable populations [20].   

Comments 11: Discussion: “This section needs to be expanded slightly in two domains:  b) Comparisons with similar studies in other relevant populations, or with general population studies of youth. Even if this is the first study to specifically examine mood disorders, it would be interesting to contrast the current findings with those seen in young adults in general (e.g., Wheeler et al., 2020; Nguyen et al., 2023) or in people with substance use disorders (e.g., Kudrich et al., 2024).”

Response 11: We agree. In response, we have modified the discussion to more clearly discuss the study findings in comparison to other studies and populations including US young adults from the general population and people with substance use disorders.  As part of this revision, we have also added several new citations to the manuscript including the one’s referenced by this reviewer (Wheeler et al., 2020; Nguyen et al., 2023; Kudrich et al., 2024).   

The updated Discussion section is presented below with changes depicted in bold:

Updated Discussion Section (Pages 15-16, Lines 438-493):

“Results from this study show that US youth receiving mood disorder treatment and their parents perceive cannabis and CBD products to be safe and effective treatments for multiple pediatric mental health conditions. Most youth and parents in the sample reported believing that regular use of cannabis and CBD products reduce depression and anxiety, with nearly half of youth and one-third of parent respondents believing cannabis use to be beneficial for suicidal thoughts and behaviors, in direct contrast to the current evidence [9, 38].  Additionally, one-in-five youth respondents reported believing that cannabis is beneficial in treating psychosis, which is concerning in light of research showing links between cannabis use and psychosis [18]. Our results are consistent with previous studies showing trends towards more positive cannabis related attitudes and medical beliefs, elevated perception of health benefits, reduced perception of risk for harm, and increased use rates across the entire US adult population and subgroups of US adolescents and young adults [20, 22-24, 26].  Additionally, our youth findings are consistent with prior studies on CBD attitudes in young adults and individuals with substance use disorders.  Our parent findings align with a recent study by Schwaller and colleagues that found positive safety perceptions about unlicensed CBD products and high administration rates of these products to children with ADHD, autism spectrum disorder, and anxiety disorder by US parents who had a 7-18 year-old child diagnosed with those conditions [54].  Our attitudinal results also parallel findings from international studies which have shown shifting societal attitudes, policies, and use patterns related to both cannabis and CBD in many countries around the world. Our study findings build off of national survey studies showing that trends towards more favorable attitudes and perceptions about cannabinoids are also occurring in vulnerable subgroups, in this case, youth with mood disorders.  These findings are of particular importance, given the current mental health crisis in US youth and the US mental healthcare system’s struggle to respond to this crisis, made more complicated by the rapid growth of multibillion-dollar cannabis and CBD product markets which have promoted broad claims of health benefits unsupported by the scientific literature [3, 39].  Recent meta-analyses and longitudinal studies show that cannabis use in adolescents with and without mood disorders is associated with poorer mood outcomes, reduced treatment response in those receiving mental health care and increased risk for depression, suicidal ideations, and suicide attempts by young adulthood [9, 38]. Concerningly, regular cannabis use and cannabis use disorders may increase mortality in young people with mood disorders.  A recent study by Fontanella et al. (2021) that examined Ohio Medicaid Claims data from 204,780 US youth (ages 10-24) found that, among youth with mood disorders, having a cannabis use disorder diagnosis was associated with increased non-fatal self-harm and all-cause mortality.  In contrast to the evidence for harm from cannabis use, the evidence for benefit from cannabinoid-based therapeutics for psychiatric conditions remains in its infancy. Safety and efficacy data on CBD (and other cannabinoids) as therapeutic agents for mood disorders remains limited, especially in youth samples; although early studies have shown promise for select conditions (e.g., anxiety), and multiple clinical trials are underway. A recent systematic review by Hasbi et al. (2023) examined the therapeutic role of cannabinoids including THC and CBD for psychiatric conditions and concluded that the evidence that cannabinoids improve depression or anxiety disorders is currently weak and of low-quality [40].  In sum, there is currently moderate to strong scientific evidence showing elevated risk for negative health outcomes from cannabis exposure in youth with and without mood disorders and insufficient evidence for safety and efficacy of cannabis or CBD products for pediatric neuropsychiatric conditions beyond rare treatment resistant seizure disorders.  Taken within this context, our attitudinal findings point to low cannabinoid health literacy and significant discrepancies between what youth with mood disorders and their parents believe about the risks and benefits of cannabis and CBD use and the current evidence related to safety and efficacy of these products, particularly in relation to cannabis’s effects on mental health and the therapeutic use of CBD and other cannabinoids for the treatment of mood disorders, psychotic disorders, and suicidal behaviors.”

Comments 12: Conclusions: “Were any participants aware of the potential risks / harms associated with cannabis (not CBD) use in mood disorders? This could be highlighted after lines 122-123.”

Response 12:  We have modified the Conclusions in response to reviewer 2’s comment 12.  The revised sentence of the conclusion (pg. 18, lines 616-618) now reads as follows (with changes depicted in bold):

“This study found that US youth receiving mood disorder treatment and their parents perceive cannabis and CBD to be safe and effective for mental health problems and to have low risk for harm from regular useFurther, it identified factors that increase the likelihood of future cannabis use in youth.”   

4. Response to Comments on the Quality of English Language

No comments made on the quality of English language

Response: N/A

5. Additional clarifications

N/A

Reviewer 3 Report

Comments and Suggestions for Authors

I read with interest the manuscript "Differences in cannabis and cannabidiol related attitudes, perceptions, and behaviors between U.S. adolescents receiving mood disorder treatment and their parents across states with and without cannabis legalization". The topic is highly relevant given ongoing changes in cannabis legislation and the vulnerability of youth with mood disorders. 

1. The title is overly long and could be more concise while retaining its meaning—for instance, "Cannabis and CBD Attitudes in Adolescents with Mood Disorders and Their Parents Across U.S. Legal Contexts."

2. The survey instrument (MABS) is not psychometrically validated. While items are based on existing instruments, the lack of formal validation should be more clearly acknowledged as a limitation in the Discussion.

3. The Methods section could be more structured. A clearer separation and more detail under subheadings—such as participant recruitment, data collection tools, and statistical analyses—would improve readability and replicability.

4. The finding that youth in recreational cannabis law (RCL) states exhibit more favorable cannabis attitudes is important. However, conclusions about causality should be made with caution, given the cross-sectional design.

5. The absence of data on race and ethnicity is a notable limitation, particularly given known disparities in cannabis-related legal and health outcomes. This point should be expanded in the limitations section.

Author Response

Manuscript ID#: IJERPH-3801705

Corresponding Author: Christopher J. Hammond

Author Responses to Reviewer 3 Comments

Response to Reviewer 3 Comments

1. Summary

Thank you for your detailed and thoughtful feedback on our manuscript. Please find the detailed responses below and the corresponding revisions/corrections highlighted/in track changes in the re-submitted files.

2. Questions for General Evaluation

Reviewer’s Evaluation

Response and Revisions

Does the introduction provide sufficient background and include all relevant references?

Can be improved

See revisions to introduction section on pgs. 2-3.

Are all the cited references relevant to the research?

Yes

Is the research design appropriate?

Can be improved

See revisions to methods section on pgs. 3-6.

Are the methods adequately described?

Can be improved

See revisions to methods section on pgs. 3-6.

Are the results clearly presented?

Can be improved

See revisions to results section on pgs. 6-14.

Are the conclusions supported by the results?

Can be improved

See revisions to discussion and conclusion sections on pgs. 14-18.

Are all figures and tables clear and well presented?

Can be improved

See revisions to tables.

3. Point-by-point response to Comments and Suggestions for Authors

Comments 1: “I read with interest the manuscript… The topic is highly relevant given ongoing changes in cannabis legislation and the vulnerability of youth with mood disorders.”

Response 1: Thank you for your interest in our work and thoughtful comments on this manuscript.

Comments 2: Title: “The title is overly long and could be more concise while retaining its meaning—for instance, "Cannabis and CBD Attitudes in Adolescents with Mood Disorders and Their Parents Across U.S. Legal Contexts.”

Response 2: Thank you for this feedback. You and reviewer 4 both commented on the title being too long.  In response, we have shortened the title.  Incorporating your and reviewer 4’s feedback, the revised title now reads as follows: “Differences in cannabis and cannabidiol attitudes, perceptions, and behaviors between U.S. adolescents receiving mood disorder treatment and their parents across legal contexts”.  We are happy to shorten further at the reviewer and editors discretion.

Comments 3: Survey Instrument: “The survey instrument (MABS) is not psychometrically validated. While items are based on existing instruments, the lack of formal validation should be more clearly acknowledged as a limitation in the Discussion.”

Response 3: Thank you for your comment. This comment about providing additional information about the survey instrument and its properties aligns with other reviewer comments (see reviewer 1 comment 4 and reviewer 2 comment 6 and author responses to each). Taking your and other reviewer comments in aggregate, we agree with the overarching recommendation to update the manuscript to improve clarity about the survey instrument and its properties and have done so in the revised manuscript.

In response to your comment we have revised the discussion section to include the lack of formal validation of the full instrument as a study limitation. 

The updated section of text in the manuscript in response to your comment is presented below with changes depicted in bold:

Updated Discussion Section (Page 17, Lines 575-584):

One study limitation is related to measurement validation and content. While we used a formative content validation approach to develop the survey and included items from validated instruments, the full survey has not been psychometrically validated...” 

Comments 4: Methods: “The Methods section could be more structured. A clearer separation and more detail under subheadings—such as participant recruitment, data collection tools, and statistical analyses—would improve readability and replicability.”

Response 4: Thank you for pointing this out. We agree.  In response we have restructured the methods section using subheadings more judiciously.  We feel that these changes have made the Methods section more clear and readable. 

The updated parts of the Methods section are presented below with changes depicted in bold:

Updated Methods Section (Pages 3-6, Lines 143-276):

2. Materials and Methods

2.1. Study Overview

              This report used youth and parent data from an anonymous online survey developed and administered to youth receiving mood disorder treatment in the US and their parents and mental health providers as part of a larger multisite cross-sectional study examining cannabis and CBD related attitudes, beliefs, and behaviors across states with different CLs. Research procedures were approved by each site’s Institutional Review Board (IRB). Strengthening the Reporting of Observational Studies in Epidemiology (STROBE) reporting guideline for cross-sectional studies were used [30].

2.2 Participants and Procedures

2.2.1. Participants

    Adolescents (ages 12-17) and young adults (ages 18-25) receiving treatment at youth mood disorder clinics in six states and parents/caregivers (henceforth referred to as parents) of youth receiving mood disorder treatment were invited to complete an anonymous survey about cannabis and CBD related attitudes and behaviors.

2.2.2. Study Eligibility

   For youth, inclusion criteria included ages 12-25 years and currently receiving mental health treatment at one of the six participating youth mood disorder clinics. For parents, inclusion criteria included being a primary caregiver of an adolescent who was currently receiving mental health treatment at the clinics. Exclusion criteria for both respondent groups included lack of English language fluency and cognitive deficits or acute severe mental illness that would interfere with their ability to safety participate in study procedures, understand item content, and complete the survey. Parent and youth participants were not limited to parent-child dyads. Broad inclusion criteria and few exclusion criteria were used to optimize generalizability by allowing participation by a diverse patient population representative of real-world patients from youth mood clinic settings. 

2.2.3. Recruitment

    Participants were recruited from NNDC affiliated youth mood disorder clinics in six states with variable state-level CLs.  Sites were geographically distributed, included urban and rural populations, and represented US Mid-Atlantic, North Central, and Mountain regions.

2.2.4. Procedures

    Consent was obtained from participants ≥18 years, parental consent and assent was obtained from participants < 18 years. Recruitment and study procedures were the same across sites, with data collection conducted by local study team members. Participation was optional and a $10 gift card was provided as compensation.  Surveys were administered online using REDCap software; one site used paper surveys. Data collection occurred July 2020 to March 2023.

2.3. Measures

2.3.1. Marijuana Attitudes Beliefs and Behaviors Study (MABS) survey instrument  

     The MABS survey was developed for this study from themes elicited from group discussions of child mood disorder experts at NNDC Child and Adolescent Mood Disorder Interest Group meetings focused around better understanding youth, parent, and provider attitudes, beliefs, and behaviors related to cannabis and CBD in the current US healthcare environment and cannabis policy landscape. Survey development was guided by expert discussion around clinically relevant themes for youth mood disorders and informed by a scoping literature review on the topic with items chosen or developed to address existing gaps in the literature. Many survey items were modeled after those used in previous survey studies [28, 31-33] with some taken directly from validated instruments (e.g., risk perception items). The full survey included 129 items querying cannabis and CBD related attitudes and beliefs (5 items), use intentions (3 items), health perception of risk for harm and medical benefit (4-items), perceived accessibility (1 item), perceived use norms (1 item), perceived knowledge about cannabis and CBD use effects (12 items), MC and CBD use behaviors (4 items), expectancies about cannabis, MC, and CBDs’ effects on mood, anxiety, and other psychiatric symptom domains (48-items), along with parent-youth communication and parenting practices (31 items), mental health provider practice behaviors and patient-provider communication about cannabis, MC, and CBD (11 items), demographics (5 items) and clinical characteristics (6 items). The survey used branching logic. After completing demographic and clinical questions, participants receive definitions for MC and CBD and were asked if they had heard of these products.  Participants who had not heard of MC or CBD were instructed to skip related survey items. Parallel youth, parent, and provider versions of the survey were created. See Supplemental methods S1-2 for additional details.

     Items from the MABS survey used as variables in the current report included: demographics (age, sex, parent education, state of residence), clinical/treatment characteristics (lifetime psychiatric symptoms from different domains/categories, current treatment type/setting/modality), cannabis and CBD medical beliefs and attitudes, perceptions about risk for harm and benefits for mental health from regular use of cannabis and CBD, cannabis and CBD use intentions and behaviors.

2.3.2. Marijuana Effect Expectancy Questionnaire-Brief (MEEQ-B)

    The MEEQ-B, a 6-item validated questionnaire, measures individual cannabis expectancies across six domains (cognitive and behavioral impairment, relaxation/tension reduction, social facilitation, sexual facilitation, cognitive/perceptual enhancement, global negative effects, and craving/physical effects). It includes positive and negative effect expectancy subscales and has been validated in adolescent and adult samples[33].

2.3.3. Patient Health Questionnaire-2-item version Depression scale (PHQ-2)

    The PHQ-2 [34] is a brief depression measure that includes the first two items from the PHQ-9 and assesses the frequency of depressive symptoms in the past 2 weeks. It has demonstrated validity in adolescent populations[35]. PHQ-2 scores range from 0-6 with a cutoff of > 3 representing an optimally sensitive/specific cutpoint for identifying individuals with a current depressive disorder.

2.3.4. Generalized Anxiety Disorder-2-item Anxiety scale (GAD-2)

     The GAD-2 is a brief anxiety measure that includes the first two items from the GAD-7 and assesses the frequency of anxiety symptoms in the past 2 weeks [36]. It has demonstrated validity in adolescent populations [37].  GAD-2 scores range from 0-6 with a cutoff of > 3 representing an optimally sensitive/specific cutpoint for identifying individuals with a current anxiety disorder.

    MEEQ-B positive and negative expectancy scores and PHQ-2, and GAD-2 scores were used as continuous measures of cannabis expectancy, depression, and anxiety in the analyses.

2.4. State-level cannabis laws

    Using US National Conference of State Legislatures data [1], respondent-reported state of residence was linked with state-level CL, and coded into one-of-three hierarchical categories: (1) RCL; (2) MCL; and (3) No-CL. All RCL states previously enacted MCL, which remained active; no states changed CL status during data collection. See supplemental eTables S1-S2.

2.5. Statistical analysis 

    We used IBM SPSS Statistical Analysis Software v25.0 (IBM, Armonk, NY); significance was set at two-sided p<0.05. Discrete missing values were minimal (0% to 3.0%) and imputed using mean substitution methods. Results were summarized using descriptive statistics to characterize beliefs, attitudes, perceptions, and expectancies, for cannabis and CBD in the total sample and across respondent groups. Multinomial logistic and linear regression analyses with state-level CL status (RCL vs. MCL vs. no-CL) and respondent group (youth vs. parent) as between-group factors tested for state-level CL and intergenerational differences. We ran separate post-hoc covariate adjusted models for group and CL status effects in the total sample, then examined CL status effects separately for youth and parents to parse group-by-CL-status interactions. Age, sex, parent education, and site were included as covariates of no interest in the models. We used two-step analysis to investigate demographic, clinical, attitudinal, and state CL factors that were associated with intentions to use cannabis and CBD in the youth respondent group.  As a first step, to reduce data, we ran ordinal regression models for each factor, covarying age, sex, parent education, and site. Then, we used multivariable ordinal regression analyses to characterize predictors of use intentions, incorporating each factor that significantly correlated with cannabis and CBD use intentions to determine whether those factors were associated with intentions to use after controlling for other relevant variables.

2.6. Sensitivity analyses

    Four sensitivity analyses were performed to ensure robustness of main results and to test whether differences related to group and CL status varied as a function of other measured and unmeasured factors. First, we applied leave-one-out validation (LOOV), holding out/censoring respondents from each site and running main analytic models in the n - 1 sample to ensure results were not being driven by a single site. Second, we reran analyses in a subgroup of matched youth-parent dyads (30 respondents/group). This test enabled us to examine differences in beliefs, attitudes, perceptions, and expectancies in relation to respondent group and CL status in a sample with limited between-group variation from measured and unmeasured environmental factors.  Third, we reran analyses in youth with current depression and anxiety disorders (i.e., PHQ-2 or GAD-2 >3) to determine impact of current depression or anxiety meeting clinical/diagnostic thresholds influenced the associations identified in our main analyses. Finally, we reran analyses for youth with no lifetime history of alcohol or drug use problems (75 youth, 57 parent report).”      

Comments 5: Discussion: “The finding that youth in recreational cannabis law (RCL) states exhibit more favorable cannabis attitudes is important. However, conclusions about causality should be made with caution, given the cross-sectional design.”

Response 5: In response we have revised the Discussion section to highlight this finding and emphasize its importance while couching our interpretation cautiously given the cross sectional design of the study.      

The updated sections of text in the manuscript are presented below with changes depicted in bold:

Updated Discussion Section (Page 16, Lines 527-534):

“Our findings showing more favorable cannabis and CBD attitudes among youth with mood disorders and parents who live in RCL states may have public health relevance. At the same time, it is important to interpret these findings cautiously given the sample size and cross-sectional design of the study which precludes causal or directional determination. Future research is needed to replicate this work and further clarify the impact of changing cannabis policies on attitudes, perceptions, and behaviors in vulnerable populations, and should be used to guide the development of targeted cannabis prevention and educational messaging for these groups.”

Comments 6: Discussion: “The absence of data on race and ethnicity is a notable limitation, particularly given known disparities in cannabis-related legal and health outcomes. This point should be expanded in the limitations section.”

Response 6: We agree. In response, we have highlighted this in the limitations subsection, expanded our discussion on race and ethnicity differences and health disparities in cannabis-related legal and health outcomes, and added new references.

The updated sections of text in the manuscript are presented below with changes depicted in bold:

Updated Discussion Section (Page 18, Lines 591-597):

One study limitation is related to measurement validation and content…. The absence of race and ethnicity data is another notable limitation of this study.  Especially given evidence for racial and ethnic differences in cannabis use [80,81] and negative health outcomes related to cannabis use [82], along with racial/ethnic disparities in cannabis use disorder treatment access and cannabis-related arrests that disproportionately impact black and indigenous people of color [82,83].  Future research should test for racial and ethnic differences and examine the role of minority stress on cannabis-related attitudes, use intentions, and behaviors.”

4. Response to Comments on the Quality of English Language

No comments made on the quality of English language

Response: N/A

5. Additional clarifications

N/A

Reviewer 4 Report

Comments and Suggestions for Authors

Dear Authors,

I was pleased to read your article, which addresses a topic of great current relevance and interest. The methodological framework appears solid and well-structured; however, the discussion is too weak as it does not adequately relate your results to the existing literature. Below are my comments and suggestions:

Revision

Title 

Too long. I would suggest removing “across states with and without cannabis legalization” and simply replacing it with “in USA”.

Introduction 

The topic of "legalization" is also relevant in sports. WADA modified the permitted thresholds for THC in athletes' samples in 2013 and subsequently excluded CBD from the Prohibited List, thus permitting its use. Furthermore, it should be considered that legal cannabis-based products may contain traces of THC. You could integrate these observations by citing the article by Ruggiero et al. (Sports, 2025): “Trends in Antidepressant, Anxiolytic, and Cannabinoid Use Among Italian Elite Athletes (2011–2023): A Longitudinal Anti-Doping Analysis”.

Materials and Methods

  • In the paragraph headings, write the full name of the questionnaires first, followed by the acronym in parentheses.
  • Paragraph 2.2.1 is well described and refers to the supplement. In the other paragraphs, however, the description is too synthetic (two lines). It would be appropriate to expand them briefly, following the model of 2.2.1.
  • Add the indication of approval by the ethics committee.

Results

  • Table 1: In the note, add a line break for the three references (“ATotal psychiatric problem score…”, “Bcurrent depressive disorder diagnosis…”, “Ccurrent anxiety disorder diagnosis…”). Replace the "statistic" column with an indication of the statistical test used.
  • Starting from page 10, the line numbering on the right starts over.

Discussion

This section requires substantial strengthening. It currently lacks a real comparison with the international and national literature. Some questions that could guide you in the integration:

  • Are there other international studies on the perception of cannabis?
  • To what extent are the positive beliefs about cannabis/CBD in young people with mood disorders in line with or in contrast to the clinical evidence on adverse effects?
  • What literature supports or refutes the idea that psychiatric patients are more vulnerable to social messages about the alleged therapeutic properties of cannabis?
  • How do your data compare with those from national surveys on the general population?
  • Are there studies documenting the effectiveness of educational interventions aimed at parents and children on the topic of cannabis?
  • What is the perception and regulation of these substances in the world of sports?

Bibliography

It does not follow the journal's editorial guidelines and must be adapted.

With these improvements, I believe the article could be ready for publication.

Best regards

Author Response

Manuscript ID#: IJERPH-3801705

Corresponding Author: Christopher J. Hammond

Author Responses to Reviewer 4 Comments

Response to Reviewer 4 Comments

1. Summary

Thank you for taking the time to provide thoughtful feedback on our manuscript. Please find the detailed responses below and the corresponding revisions/corrections highlighted/in track changes in the re-submitted files.

2. Questions for General Evaluation

Reviewer’s Evaluation

Response and Revisions

Does the introduction provide sufficient background and include all relevant references?

Yes

Are all the cited references relevant to the research?

Yes

Is the research design appropriate?

Yes

Are the methods adequately described?

Can be improved

See revisions to method section on pgs. 3-6.

Are the results clearly presented?

Must be improved

See revisions to results section on pgs. 6-14.

Are the conclusions supported by the results?

Must be improved

See revisions to discussion and conclusions sections on pgs. 14-18.

Are all figures and tables clear and well presented?

Can be improved

See revisions to tables.

3. Point-by-point response to Comments and Suggestions for Authors

Comments 1: “I was pleased to read your article, which addresses a topic of great current relevance and interest. The methodological framework appears solid and well-structured…”

Response 1: Thank you for your detailed review and thoughtful feedback on our manuscript.  In particular, we appreciate the comments noted above that reference the current relevance of the topic of the manuscript and the solid and well-structured methodological framework used. 

Comments 2: Title: “Too long. I would suggest removing “across states with and without cannabis legalization” and simply replacing it with “in USA”.

Response 2: Thank you for this feedback. You and reviewer 3 both commented on the title being too long.  In response, we have shortened the title.  Incorporating your and reviewer 3’s feedback, the revised title now reads as follows: “Differences in cannabis and cannabidiol attitudes, perceptions, and behaviors between U.S. adolescents receiving mood disorder treatment and their parents across legal contexts”

Comments 3: Introduction: “The topic of "legalization" is also relevant in sports. WADA modified the permitted thresholds for THC in athletes' samples in 2013 and subsequently excluded CBD from the Prohibited List, thus permitting its use. Furthermore, it should be considered that legal cannabis-based products may contain traces of THC. You could integrate these observations by citing the article by Ruggiero et al. (Sports, 2025): “Trends in Antidepressant, Anxiolytic, and Cannabinoid Use Among Italian Elite Athletes (2011–2023): A Longitudinal Anti-Doping Analysis.”

Response 3:  Thank you for your thoughtful feedback about the introduction. We agree that the introduction would benefit from revisions to present more context about legal and regulatory changes related to cannabinoids including THC and CBD.  However, we also feel that expanding to discuss legalization in relation to sports is beyond the scope of the current manuscript (and warrants its own thematic articles).  In order to be responsive to reviewer 4 comment 3, we have revised the introduction significantly to more clearly discuss cannabis legalization, distinctions between THC and CBD, and the relevance for our sample population of youth with mood disorders and their parents.  While we did not expand the introduction to include a broad description about sports, we did mention briefly the impact on changes in THC and CBD legalization on sports and incorporated the Ruggerio et al. 2025 citation into the manuscript (pg 3 lines 108-111) in the following sentence:

 “Furthermore, several studies have shown that commonly available hemp-derived CBD supplements are often mislabeled, contaminated, or adulterated, and may contain trace amounts of THC in sufficient doses to produce psychoactive effects or result in positive urine drug test results for cannabis [27] that could impact health, employment status, and sports participation [28].”

Comments 4: Methods: “In the paragraph headings, write the full name of the questionnaires first, followed by the acronym in parentheses.”

Response 4:  The authors apologize for our confusion with regard to this comment and are asking for additional clarification so we can be appropriately responsive.  Reviewer 4’s comment 4 states “In the paragraph headings, write the full name of the questionnaires first, followed by the acronym in parentheses.”  Perhaps, we are misunderstanding what reviewer 4 is asking for us to change. As we believe that what reviewer is asking for is how the questionnaires are presented currently in the methods section.  We have presented the questionnaires and their acronyms as they are shown in the Methods section of the manuscript below.  Please let us know if you would like us to change from this to another format.  We do not feel strongly about this and defer to the reviewer and editors in this regard.   

Relevant parts of the Methods Section (pgs 4-5 lines 185-236) related to this comment are shown below

2.3.2. Marijuana Effect Expectancy Questionnaire-Brief (MEEQ-B)

    The MEEQ-B, a 6-item validated questionnaire, measures individual cannabis expectancies across six domains (cognitive and behavioral impairment, relaxation/tension reduction, social facilitation, sexual facilitation, cognitive/perceptual enhancement, global negative effects, and craving/physical effects). It includes positive and negative effect expectancy subscales and has been validated in adolescent and adult samples[33].

2.3.3. Patient Health Questionnaire-2-item version Depression scale (PHQ-2)

    The PHQ-2 [34] is a brief depression measure that includes the first two items from the PHQ-9 and assesses the frequency of depressive symptoms in the past 2 weeks. It has demonstrated validity in adolescent populations[35]. PHQ-2 scores range from 0-6 with a cutoff of > 3 representing an optimally sensitive/specific cutpoint for identifying individuals with a current depressive disorder.

2.3.4. Generalized Anxiety Disorder-2-item Anxiety scale (GAD-2)

     The GAD-2 is a brief anxiety measure that includes the first two items from the GAD-7 and assesses the frequency of anxiety symptoms in the past 2 weeks [36]. It has demonstrated validity in adolescent populations [37].  GAD-2 scores range from 0-6 with a cutoff of > 3 representing an optimally sensitive/specific cutpoint for identifying individuals with a current anxiety disorder.

    MEEQ-B positive and negative expectancy scores and PHQ-2, and GAD-2 scores were used as continuous measures of cannabis expectancy, depression, and anxiety in the analyses.   

Comments 5: Methods: “Paragraph 2.2.1 is well described and refers to the supplement. In the other paragraphs, however, the description is too synthetic (two lines). It would be appropriate to expand them briefly, following the model of 2.2.1”

Response 5: Thank you for this feedback.  We agree, and have revised the Methods section, expanding the other sections as recommended by reviewer 4, following the model of 2.2.1.     

The updated sections of text in the Methods section (pgs 4-5, lines 185-236) related to this comment are presented above in our response to reviewer 4 comment 4 (with changes depicted in bold).

Comments 6: Methods: “Add the indication of approval by the ethics committee”.

Response 6:  In response we re-reviewed the manuscript.  This information is currently listed in the methods section (pg. 4, lines 148-149) where we state, “Research procedures were approved by each site’s Institutional Review Board (IRB).”  We have also included expanded information about this in the Institutional Review Board Statement at the end of the manuscript (pg. 19, lines 663-668, shown below).  Please let us know if anything is missing or if there is additional information needed regarding our description of the study’s ethics committee/IRB approval.   

“Institutional Review Board Statement: The study was conducted in accordance with the Declaration of Helsinki, and approved by the Institutional Review Board of Johns Hopkins University School of Medicine (IRB00221389). Research procedures were also approved by each site’s Institutional Review Board (Johns Hopkins University, University of Colorado, Mayo Clinic, University of Illinois Chicago, Indiana University, University of Michigan). Johns Hopkins University served as the coordinating and data analysis center for the study.”

Comments 7: Results: Table 1: “In the note, add a line break for the three references (“ATotal psychiatric problem score…”, “Bcurrent depressive disorder diagnosis…”, “Ccurrent anxiety disorder diagnosis…”). Replace the "statistic" column with an indication of the statistical test used.”

Response 7:  In response we have added a line break for the three referenced footnotes in the table. At the reviewer’s suggestion, in the statistics column, we have shown which statistical test was used for each analysis. Of note, in the footnote of the table, we have included a sentence that reads, “Group comparisons were made using independent t-tests and chi-squares for continuous and categorical variables, respectively.”  The revised table 1 is shown on pgs. 7-8.       

Comments 8: Results: “Starting from page 10, the line numbering on the right starts over.”

Response 8: We have fixed the line numbering error noted by the reviewer so that the line numbering is continuous from the beginning.

Comments 9: Discussion: “The discussion is too weak as it does not adequately relate your results to the existing literature” and “This section requires substantial strengthening. It currently lacks a real comparison with the international and national literature.” Some questions to guide you in integration:

•           Are there other international studies on the perception of cannabis?

•           To what extent are the positive beliefs about cannabis/CBD in young people with mood disorders in line with or in contrast to the clinical evidence on adverse effects?

•           What literature supports or refutes the idea that psychiatric patients are more vulnerable to social messages about the alleged therapeutic properties of cannabis?

•           How do your data compare with those from national surveys on the general population?

•           Are there studies documenting the effectiveness of educational interventions aimed at parents and children on the topic of cannabis?

•           What is the perception and regulation of these substances in the world of sports?

Response 9: Thank you for your feedback on our discussion and guiding questions around how to modify/improve it.  In response we have made major edits to our discussion section to discuss the findings within the context of the current scientific literature (US and international).  We have used your thoughtful questions to assist with these changes. As part of these revisions we have also added several citations.  

The updated sections of text in the manuscript are presented below with changes depicted in bold:

Updated Discussion Section (pgs 15-17, Lines 438-572):

“Results from this study show that US youth receiving mood disorder treatment and their parents perceive cannabis and CBD products to be safe and effective treatments for multiple pediatric mental health conditions. Most youth and parents in the sample reported believing that regular use of cannabis and CBD products reduce depression and anxiety, with nearly half of youth and one-third of parent respondents believing cannabis use to be beneficial for suicidal thoughts and behaviors, in direct contrast to the current evidence [9, 38].  Additionally, one-in-five youth respondents reported believing that cannabis is beneficial in treating psychosis, which is concerning in light of research showing links between cannabis use and psychosis [18]. Our results are consistent with previous studies showing trends towards more positive cannabis related attitudes and medical beliefs, elevated perception of health benefits, reduced perception of risk for harm, and increased use rates across the entire US adult population and subgroups of US adolescents and young adults [20, 22-24, 26].  Additionally, our youth findings are consistent with prior studies on CBD attitudes in young adults and individuals with substance use disorders.  Our parent findings align with a recent study by Schwaller and colleagues that found positive safety perceptions about unlicensed CBD products and high administration rates of these products for children with ADHD, autism spectrum disorder, and anxiety disorder by US parents who had a 7-18 year-old child diagnosed with those conditions [54].  Our attitudinal results also parallel findings from international studies which have shown shifting societal attitudes, policies, and use patterns related to both cannabis and CBD in many countries around the world. Our study findings build off of national survey studies showing that trends towards more favorable attitudes and perceptions about cannabinoids are also occurring in vulnerable subgroups, in this case, youth with mood disorders.  These findings are of particular importance, given the current mental health crisis in US youth and the US mental healthcare system’s struggle to respond to this crisis, made more complicated by the rapid growth of multibillion-dollar cannabis and CBD product markets which have promoted broad claims of health benefits unsupported by the scientific literature [3, 39].  Recent meta-analyses and longitudinal studies show that cannabis use in adolescents with and without mood disorders is associated with poorer mood outcomes, reduced treatment response in those receiving mental health care and increased risk for depression, suicidal ideations, and suicide attempts by young adulthood [9, 38]. Concerningly, regular cannabis use and cannabis use disorders may increase mortality in young people with mood disorders.  A recent study by Fontanella et al. (2021) that examined Ohio Medicaid Claims data from 204,780 US youth (ages 10-24) found that, among youth with mood disorders, having a cannabis use disorder diagnosis was associated with increased non-fatal self-harm and all-cause mortality.  In contrast to the evidence for harm from cannabis use, the evidence for benefit from cannabinoid-based therapeutics for psychiatric conditions remains in its infancy. Safety and efficacy data on CBD (and other cannabinoids) as therapeutic agents for mood disorders remains limited, especially in youth samples; although early studies have shown promise for select conditions (e.g., anxiety), and multiple clinical trials are underway. A recent systematic review by Hasbi et al. (2023) examined the therapeutic role of cannabinoids including THC and CBD for psychiatric conditions and concluded that the evidence that cannabinoids improve depression or anxiety disorders is currently weak and of low-quality [40].  In sum, there is currently moderate to strong scientific evidence showing elevated risk for negative health outcomes from cannabis exposure in youth with and without mood disorders and insufficient evidence for safety and efficacy of cannabis or CBD products for pediatric neuropsychiatric conditions beyond rare treatment resistant seizure disorders.  Taken within this context, our attitudinal findings point to low cannabinoid health literacy and significant discrepancies between what youth with mood disorders and their parents believe about the risks and benefits of cannabis and CBD use and the current evidence related to safety and efficacy of these products, particularly in relation to cannabis’s effects on mental health and the therapeutic use of CBD and other cannabinoids for the treatment of mood disorders, psychotic disorders, and suicidal behaviors.

    We also found evidence of intergenerational differences in cannabis and CBD related beliefs, attitudes, and perceptions between youth and parent groups that remained significant after controlling for different confounding variables and were robust to sensitivity tests.  Consistent with prior studies in general population samples, our results showed that youth with mood disorders were significantly more likely to report believing that cannabis is safe and effective for pediatric mental health conditions and to have lower perception of risk for harm related to youth cannabis use, and greater perception of health benefit related to youth cannabis and CBD use than parents [28].  Intergenerational differences in attitudes, beliefs, and perceptions seen in this study were observed across all cannabinoid product types, chemotypes, THC/CBD potencies, use frequencies, and methods of administration. This suggests a possible class effect for cannabinoids by age/generational group.  Group differences in perceptions and attitudes could also be explained by developmental differences in sensation seeking, impulsiveness, and reward and risk processing between youth and adult respondents [41].

    One of the main objectives of this study was to examine the influence of state-level CL on cannabis and CBD attitudes, expectancies, health perceptions, and behaviors in youth and parent respondents.  Cannabis and CBD related medical beliefs, attitudes about medical cannabis and CBD, and health benefit and risk perceptions about cannabis and MC use varied as a function of state-level CL, with some differences between youth and parent respondents.  Interestingly, our results exhibited some evidence of a gradient effect whereby respondents in RCL states had more favorable attitudes than respondents in MCL states, who generally had more favorable attitudes than respondents in No-CL states.  Notably, parents living in RCL states had more approving attitudes of youth MC and CBD use and youth living in RCL states had greater health benefit perceptions about MC. Our CL findings are consistent with previous studies showing an impact of CL on cannabis-related attitudes, medical beliefs, and perceptions in adolescents and adults [27, 42-45], although null results have also been found [46], suggesting heterogenous associations with CL impacting some attitudes and beliefs more so than others [26].  Our study also extends this research to include CBD-related attitudes and medical beliefs in this population, with our results showing that these attitudes and beliefs also differed as a function of state CL status.  Variation between our results and those found in other studies may be related to differences in population characteristics, recruitment/sampling frame, sample size, study design, covariate selection, and analytic strategy. Our findings showing more favorable cannabis and CBD attitudes among youth with mood disorders and parents who live in RCL states may have public health relevance. At the same time, it is important to interpret these findings cautiously given the sample size and cross-sectional design of the study which precludes causal or directional determination. Future research is needed to replicate this work and further clarify the impact of changing cannabis policies on attitudes, perceptions, and behaviors in vulnerable populations, and should be used to guide the development of targeted cannabis prevention and educational messaging for these groups.

    Our results also identified sociodemographic, clinical, and attitudinal factors associated with intentions to use cannabis among youth with mood disorders.  Consistent with previous studies [31, 47-51], factors associated with cannabis use intentions among youth in our sample included older age, sex, total psychiatric problem scores, lifetime presence of certain psychiatric symptom categories (impulsivity, post-traumatic stress disorder, eating disorders, non-suicidal self-injurious behaviors, suicidal thoughts and/or behaviors), positive cannabis use expectancies, favorable cannabis attitudes, and lower risk and higher health benefit perceptions related to cannabis use.   Among these, male sex, positive cannabis expectancies, and favorable cannabis attitudes were independently associated with cannabis use intentions after adjusting for age, sex, parent education, site, and each of the other variables in our multivariate analysis.

    Based on our study findings, goals of next-stage multilevel prevention efforts should focus on cannabinoid health literacy, targeting youth and parent misperceptions, providing education about mental health risks related to cannabis use; while concurrently promoting aggressive treatment of youth mood disorders to reduce the likelihood these youth will initiate cannabis or progress to regular use. Attitudinal and mental health factors represent important and malleable intervention targets and can inform public health cannabis messaging efforts [39, 50], especially as youth living in RCL states have more favorable cannabis attitudes associated with increased intention to use.  Cannabis prevention efforts for this population should largely be driven by pediatric medical and mental health clinicians; public health campaigns should focus on cannabis health literacy by providing targeted, evidence-based education to youth and parents and encouraging fact-driven discussions between parents, youth, and providers about cannabinoids and mental health.  Standardized medical/healthcare education and workforce training for pediatric mental and medical healthcare providers about cannabinoid risks and benefits and how to screen for cannabinoid product use and advise youth/families are needed. In order to be effective, this medical/healthcare education on cannabis and cannabinoids should be included as required training content for residency/fellowship programs and continued medical education content for board certification and maintenance for psychiatrists, psychologists, pediatricians, primary care providers, and other licensed mental health clinicians.  Importantly, taking lessons from the history of alcohol and tobacco industry’s influence on medical/healthcare practice, medical education content on cannabis and cannabinoids should be unbiased and evidence-informed, and consider and weight the risks and potential benefits from different cannabinoid products.  This educational content should be prepared and disseminated by experts in cannabis prevention and early intervention research without cannabis industry influence. Furthermore, future cannabis regulatory and policy reforms should incorporate strong evidence-informed risk mitigating provisions and resources that seek to reduce cannabis use in youth and protect vulnerable populations [13].

Comments 10: Bibliography: “It [the reference section] does not follow the journal's editorial guidelines and must be adapted.”

Response 10: Thank you for noting this. We have revised the bibliography to fit the IJERPH journal citation and reference guidelines.

4. Response to Comments on the Quality of English Language

Point 1: No comments on the quality of English language were made by reviewer

Response 1: N/A

5. Additional clarifications

N/A

Round 2

Reviewer 4 Report

Comments and Suggestions for Authors

Dear Authors,

I have reviewed the revised manuscript and your responses.

The changes made are satisfactory and have effectively addressed all the points raised. The manuscript is significantly improved, and I consider it ready for publication.

Best regards